# Molecular-level study on the role of methanesulfonic acid in iodine oxoacids nucleation

Jing Li[1], Nan Wu[1], An Ning[1,*], and Xiuhui Zhang[1,*]

[1]Key Laboratory of Cluster Science, Ministry of Education of China, School of Chemistry and Chemical Engineering, Beijing Institute of Technology, Beijing, 100081, China

*Correspondence to*: A. Ning (anning@bit.edu.cn) and X.H. Zhang (zhangxiuhui@bit.edu.cn)

**Abstract.** Iodic acid ($HIO_3$) and iodous acid ($HIO_2$) have been identified to nucleate effectively by the Cosmics Leaving OUtdoor Droplets (CLOUD) experiment at CERN, yet it may be hard to explain all $HIO_3$-induced nucleation. Given the complexity of marine atmosphere, other precursors may be involved. Methanesulfonic acid (MSA), as a widespread precursor over oceans, has been proven to play a vital role in facilitating nucleation. However, its kinetic impacts on synergistic nucleation of iodine oxoacids remain unclear. Hence, we investigated the MSA-involved $HIO_3$-$HIO_2$ nucleation process at the molecular level using density functional theory (DFT) and Atmospheric Clusters Dynamic Code (ACDC). The results show that MSA can form stable molecular clusters with $HIO_3$ and $HIO_2$ jointly via hydrogen and halogen bonds, as well as electrostatic attraction after proton transfer to $HIO_2$. Thermodynamically, the MSA-involved clustering can occur nearly without free-energy barrier, following $HIO_2$-MSA binary and $HIO_3$-$HIO_2$-MSA ternary pathway. Furthermore, adding MSA significantly enhance the rate of $HIO_3$-$HIO_2$-based cluster formation, even up to $10^4$-fold at cold marine regions with rich MSA and scarce iodine, such as polar Ny-Ålesund and Marambio. Thus, the proposed more efficient $HIO_3$-$HIO_2$-MSA nucleation mechanism may provide theoretical evidence for explaining the frequent and intensive burst of marine iodine particles.

## 1 Introduction

Marine aerosol, the primary natural aerosol (O'Dowd and Leeuw, 2007), significantly impacts global climate, radiation balance, and even human health (Wang et al., 2010; Pöschl, 2005). New particle formation (NPF) is a main source of marine aerosols, which proceeds via nucleation and subsequent growth (Lee et al., 2019; Zhang, 2010; Kulmala et al., 2013; Zhang et al., 2012). And the nucleation, forming critical clusters at 1-2 nm from gaseous precursors, is the pivotal step affecting NPF (Zhang, 2010; Kulmala et al., 2013). However, the chemicals involved in nucleation and the underlying mechanisms remain poorly understood, due to technological limitations in the molecular-level analysis. Additionally, the lack of comprehensive and long-term ocean observations, further hinders our knowledge of marine NPF.

Recent field studies suggest that marine NPF events are closely related to atmospheric iodine-bearing molecules emitted by algae (Yu et al., 2019; Baccarini et al., 2020; Beck et al., 2021). During the NPF events in coastal (e.g., Mace Head and Zhejiang) (Sipilä et al., 2016; Yu et al., 2019) and polar oceans (e.g., Arctic Ocean) (Baccarini et al., 2020), the nucleation

processes are mainly driven by iodic acids ($HIO_3$). Yet in fact, the self-nucleation of $HIO_3$ alone cannot explain the observed NPF rates (Rong et al., 2020). More recently, the Cosmics Leaving OUtdoor Droplets (CLOUD) experiment at CERN has found that iodous acid ($HIO_2$) plays a key role in stabilizing $HIO_3$, enabling effective nucleation by the sequential addition of $HIO_3$ followed by $HIO_2$ (He et al., 2021). Further theoretical studies uncover that the stabilizing effect of $HIO_2$ on $HIO_3$ stems from its role as a base in clustering (Zhang et al., 2022b; Liu et al., 2023). Although the efficient nucleation of $HIO_3$ and $HIO_2$

is overall consistent with the CLOUD measurement (Zhang et al., 2022a), this mechanism does not account for all $HIO_3$-induced nucleation in the real atmosphere (Ma et al., 2023). Thus, other essential precursors in marine atmosphere might potentially affect $HIO_3$-$HIO_2$ nucleation, but which and how remain largely unexamined.

Methanesulfonic acid (MSA), as a typical marine sulfur precursor, is widespread over oceans (Saltzman et al., 1983; Read et al., 2008; Chen et al., 2012; Yan et al., 2019) with considerable atmospheric concentrations ($10^5$ – $10^8$ molec. cm$^{-3}$) (Eisele

and Tanner, 1993; Dal Maso et al., 2002; Chen et al., 2018; Yan et al., 2019). Moreover, MSA has been shown to initiate nucleation with vital atmospheric precursors, such as ammonia and amines, enhancing cluster formation (O'Dowd et al., 2002; Bork et al., 2014; Shen et al., 2019; Shen et al., 2020; Brean et al., 2021; Liu et al., 2022). Importantly, current evidence suggests that MSA can also form stable clusters with $HIO_3$ or $HIO_2$ individually, but none of the resulting binary nucleation can explain field measurements well (Ning et al., 2022; Wu et al., 2023). Despite the stabilizing effect of MSA on iodine

oxoacids, it remains unknown whether MSA can synergistically nucleate with $HIO_3$ and $HIO_2$, as well as the induced kinetic impacts on clustering. Furthermore, given the coexistence of MSA and $HIO_3$ in different marine regions (Quéléver et al., 2022; Beck et al., 2021), along with the consistent presence of $HIO_3$ and $HIO_2$ as homologous substances (Sipilä et al., 2016), the importance of the $HIO_3$-$HIO_2$-MSA nucleation mechanism may differ under distinct ambient conditions, but it remains unrevealed.

Herein, we have systematically investigated the $HIO_3$-$HIO_2$-based nucleation involved in MSA, including $(HIO_3)_x(HIO_2)_y(MSA)_z$ ($1 \leq x + y + z \leq 5$, $0 \leq z \leq 3$) clusters, by combining quantum chemical (QC) approach and Atmospheric Clusters Dynamic Code (ACDC) (McGrath et al., 2012). To probe the nature of cluster formation, the wavefunction analysis was performed to investigate the intermolecular interactions. And the Gibbs free energies of cluster formation were calculated to evaluate cluster stability. Moreover, a series of ACDC simulations were executed to delve into the influence of MSA on

nucleation rates and mechanisms under varying atmospheric conditions, such as precursor concentration and temperature.

## 2 Methods

### 2.1 Quantum Chemistry Calculations

To locate the low-lying isomers of $(HIO_3)_x(HIO_2)_y(MSA)_z$ ($1 \leq x + y + z \leq 5$, $0 \leq z \leq 3$) clusters, the multi-step conformer search was adopted here (details in Supporting Information (SI)). The resulting stable clusters with the lowest energies were

identified at $\omega$B97X-D/6-311++G(3df,3pd) (for C, H, O, and S atoms) + aug-cc-pVTZ-PP with ECP28MDF (for I atom) level of theory (Francl et al., 1982; Peterson et al., 2003), and the corresponding Cartesian coordinates were collected in Table S9.

In addition, the structures of pure-$HIO_3$, pure-$HIO_2$, pure-MSA, $HIO_3$-$HIO_2$, $HIO_3$-MSA and $HIO_2$-MSA clusters in the present study were adopted from the previous studies (Rong et al., 2020; Zhang et al., 2022b; Liu et al., 2023; Ning et al., 2022; Wu et al., 2023). All density functional theory (DFT) calculations were carried out using the Gaussian 09 package (Frisch et al.,

2009), where FineGrid and tight convergence were employed. The single-point energy was calculated at the RI-CC2/aug-cc-pVTZ (for C, H, and O atoms) + aug-cc-pV(T+d)Z (for S atom) + aug-cc-pVTZ-PP with ECP28MDF (for I atom) level of theory (Hättig and Weigend, 2000) by TURBOMOLE program (Ahlrichs et al., 1989), because of its success in fitting with the experiments (Lu et al., 2020; Kürten et al., 2018; Rong et al., 2020; Almeida et al., 2013). In the present study, the Gibbs formation free energy ($\Delta G_{ref}$, kcal mol$^{-1}$) of the $HIO_3$-$HIO_2$-MSA clusters at the reference pressure (1 atm) was calculated as:

$$\Delta G_{ref} = \Delta E_{RI\text{-}CC2} + \Delta G_{thermal}^{\omega B97X\text{-}D}, \tag{1}$$

where $\Delta E_{RI\text{-}CC2}$ is the electronic contribution and $\Delta G_{thermal}^{\omega B97X\text{-}D}$ is the thermal contribution to free energy. The $\Delta G_{ref}$ at different temperatures ($T = 258 - 298$ K) were calculated using the Shermo 2.0 code (Lu and Chen, 2021) and collected in Table S1. Further given the effect of vapor pressures of the precursor, the $\Delta G_{ref}$ was converted to $\Delta G(P_1, P_2, ..., P_n)$ (Vehkamäki, 2006) by the Eq. (2):

$$\Delta G(P_1, P_2, ..., P_n) = \Delta G_{ref} - k_B T \sum_{i=1}^{n} N_i \ln\left(\frac{P_i}{P_{ref}}\right), \tag{2}$$

where $n$ is the number of components within the cluster, $k_B$ denotes the Boltzmann constant, $T$ signifies the temperature, $N_i$ refers to the number of molecules of type $i$ in the number of components in the cluster and $P_i$ is the partial pressure of component $i$ in the vapor phase.

## 2.2 Wavefunction Analysis

To probe the binding nature within molecular clusters, wavefunction analysis was conducted using Multiwfn 3.7 (Lu and Chen, 2012). The electrostatic potential (ESP) on the van der Waals (vdW) surface was calculated to identify active interaction sites. Specifically, the negative ESP region is electron-rich, while the positive ESP region is electron-deficient, potentially leading to mutual non-covalent interactions, such as hydrogen bonds (HBs) and halogen bonds (XBs). To further quantify the bond strength, the electron density $\rho(r)$, Laplacian electron density $\nabla^2\rho(r)$ and energy density $H(r)$ at bond critical points (BCPs)

were calculated based on the atoms in molecules (AIM) theory (Lane et al., 2013).

## 2.3 Atmospheric Clusters Dynamic Simulations

To explore nucleation kinetic, the Atmospheric cluster dynamics code (ACDC) (McGrath et al., 2012) was adopted here to compute the cluster formation rates, steady-state concentrations, and formation pathways by explicit solution of the birth-death equations (Eq. (3)).

$$\frac{dc_i}{dt} = \frac{1}{2}\sum_{j<i} \beta_{j,(i-j)}\, C_j C_{(i-j)} + \sum_j \gamma_{(i+j)\to i}\, C_{i+j} - \sum_j \beta_{i,j}\, C_i C_j - \frac{1}{2}\sum_{j<i} \gamma_{i\to j}\, C_i + Q_i - S_i, \tag{3}$$

where the subscripts ($i$, $j$, $i$-$j$ and $i$+$j$) denote different clusters or monomers, $C_i$ is the concentration of cluster $i$, $\beta_{i,j}$ and $\gamma_{(i+j)\to i}$ represent the cluster collision and evaporation rate coefficient, respectively. And $Q_i$ and $S_i$ denote the external source and sink terms, respectively. The $\beta_{i,j}$ is calculated as follows:

$$\beta_{i,j} = \left(\frac{3}{4\pi}\right)^{1/6} \left(\frac{6k_B T}{m_i} + \frac{6k_B T}{m_j}\right)^{1/2} \left(V_i^{1/3} + V_j^{1/3}\right)^2, \tag{4}$$

where $m_i$ and $V_i$ represent the mass and volume of cluster $i$, respectively. And $V_i = 3/4 \times \pi \times (d_i/2)^3$, where the diameter $d_i$ of cluster $i$ is derived from the cluster volume $V_i$ calculated by Multiwfn 3.7 (Lu and Chen, 2012). The $\gamma_{(i+j)\to i}$ is calculated by the Eq. (5):

$$\gamma_{(i+j)\to i} = \beta_{i,j}\frac{P_{ref}}{k_B T}\exp\left(\frac{\Delta G_{i+j} - \Delta G_i - \Delta G_j}{k_B T}\right), \tag{5}$$

where $P_{ref}$ is the reference pressure at 1 atm, and $\Delta G$ is the formation free energy of the cluster.

In the performed ACDC simulations, all possible collision and evaporation processes, including monomer-monomer, monomer-cluster and cluster-cluster collisions, as well as decomposition of parent clusters into monomers and clusters, or into two smaller clusters, were taken into account. Additionally, whether the clusters in the simulated system are stable depends on whether the rate of collision frequencies exceeds the total evaporation rate coefficients ($\beta C/\Sigma\gamma > 1$) (Table S4). The setting of the boundary conditions of ACDC simulations are summarized in Table S3. The uncertainty analysis was considered in this
study, with details provided in Supporting Information (SI).

**3 Results and Discussion**

Here, conformational analysis was first carried out to study how MSA affects intermolecular interactions in the $HIO_3$-$HIO_2$-MSA clusters. And the thermodynamic analysis was employed to assess stability of the formed clusters. To gain insights into nucleation mechanisms, a series of ACDC simulations were executed under varying atmospheric conditions.

**3.1 Cluster Conformational Analysis**

Strong interactions among nucleation precursors are pivotal for forming stable clusters. To evaluate the binding potential of MSA with $HIO_3$ and $HIO_2$, we calculated the ESP-mapped molecular vdW surface to identify interaction sites. As illustrated in Fig. 1, MSA has a positive ESP maximum (+63.95 kcal mol[-1]) at the H atom of its -OH group, serving as a HB donor. The iodine atoms of $HIO_3$ and $HIO_2$ with positive ESP maximums (+51.90 and +45.26 kcal mol[-1]), can act as effective XB donors.
Additionally, the oxygen atoms in the S=O group (from MSA) and I=O group (from $HIO_3$ and $HIO_2$) with strong

electronegativity can act as HB or XB acceptor sites, due to the lone pair electrons. Therefore, as shown in Fig. 1(d), MSA, $HIO_3$, and $HIO_2$ have the potential to form clusters via intermolecular HBs and XBs.

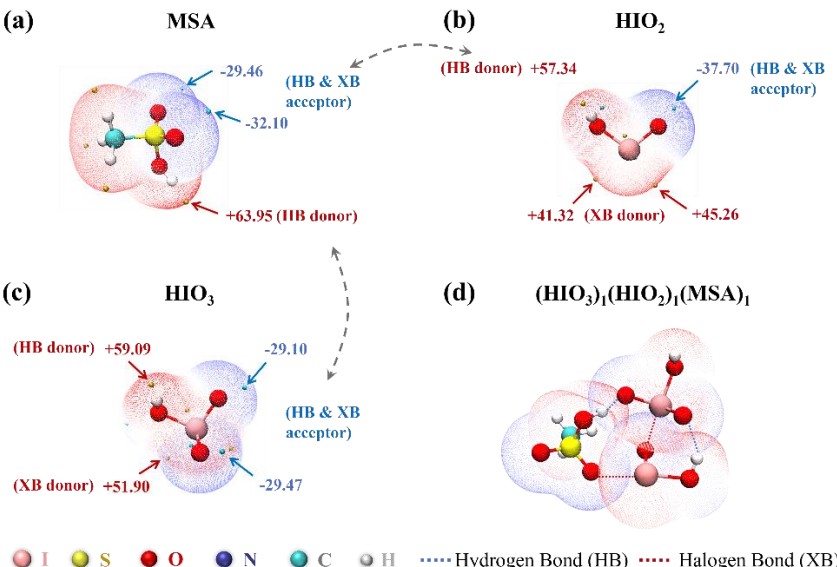

**Figure 1.** The ESP-mapped molecular vdW surface of **(a)** MSA, **(b)** $HIO_2$, **(c)** $HIO_3$ and **(d)** $(HIO_3)_1(HIO_2)_1(MSA)_1$. The golden and cyan
dots represent the positions of maximums and minimums of ESP (unit: kcal mol$^{-1}$), respectively. The gray dashed arrows signify the site-to-
site interaction tendencies.

As presented in Fig. 2, all the identified $HIO_3$-$HIO_2$-MSA clusters are structurally stabilized by the network of HBs (blue
dashed lines) and XBs (red dashed lines). Within these clusters, the inward-facing oxygen atom and hydroxyl (-OH) group in
MSA facilitates its being involved in forming more HBs and XBs, compared to the $HIO_3$-$HIO_2$ clusters (Fig. S1). Statistically,
within $HIO_3$-$HIO_2$-MSA clusters, the percentage of XBs (61%) is higher than that of HBs (39%). Notably, during the $HIO_3$-
$HIO_2$-MSA cluster formation, $HIO_2$ behaves like a base and is protonated by MSA instead of $HIO_3$, likely due to greater acidity
of MSA than $HIO_3$. After the MSA-driven proton transfer to $HIO_2$, the resulting electrostatic interactions between the formed
ion pairs ($CH_3SO_3^-$ – $H_2IO_2^+$) further stabilize the clusters. Taken together, MSA can form clusters with $HIO_3$ and $HIO_2$ via
HBs, XBs, and electrostatic attraction between ion pairs after proton transfer. Additionally, taking the $(HIO_3)_1(HIO_2)_3(MSA)_1$
cluster for example, there are still some potential remaining unoccupied binding sites as shown in Fig. S2. It suggests that the
studied large-size clusters still have unoccupied HB and XB sits that can potentially facilitate the condensation of precursors
in the atmosphere, enhancing further growth of marine aerosols.

To further quantify bond strength within $HIO_3$-$HIO_2$-MSA clusters, the topological analysis was performed based on the
atoms in molecules (AIM) theory. The electron density $\rho(r)$, Laplacian electron density $\nabla^2\rho(r)$, energy density $H(r)$ at the
corresponding bond critical points (BCPs) in the studied $HIO_3$-$HIO_2$-MSA clusters were calculated and collected in Table S2.
The $\rho(r)$ is generally positively associated with the bond strength. For the $HIO_3$-$HIO_2$-MSA clusters, $\rho(r)$ values at the BCPs

of the HBs range from 0.0090 to 0.0869 a.u., exceeding the reported threshold of HB (0.002 – 0.040 a.u.) (Koch and Popelier, 1995; Grabowski, 2004). And the associated values of $\nabla^2\rho(r)$ at these BCPs range from 0.0310 to 0.1180 a.u., falling within the suggested range for HB (0.014 – 0.139 a.u.) (Koch and Popelier, 1995; Grabowski, 2004). Meanwhile, for O–I···O XBs, the $\rho(r)$ and $\nabla^2\rho(r)$ values within HIO$_3$-HIO$_2$-MSA clusters range from 0.0068 to 0.1999 a.u. and 0.0288 to 0.1744 a.u., respectively. Collectively, MSA can stabilize HIO$_3$-HIO$_2$ clusters via more relatively strong HBs and XBs, while also protonating HIO$_2$ to form ion pairs.

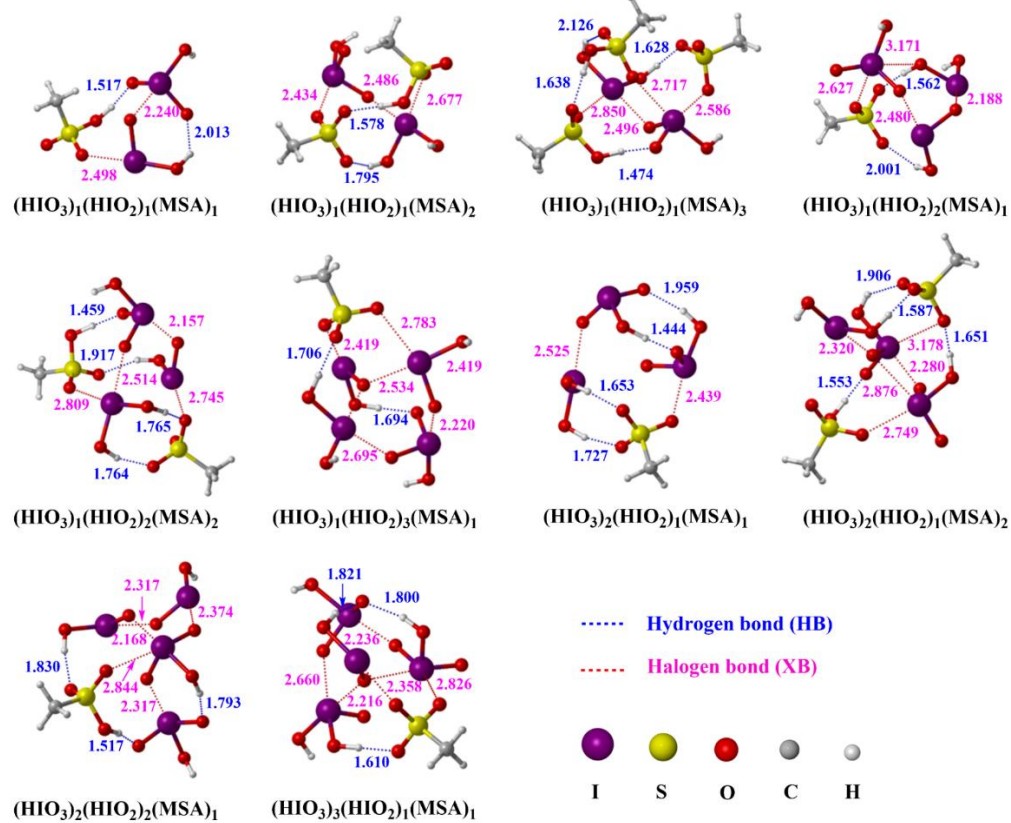

**Figure 2.** The most stable configurations of the HIO$_3$-HIO$_2$-MSA ternary clusters identified at the ωB97XD/6-311++G(3df, 3pd) (for C, H, O, and S atoms) + aug-cc-pVTZ-PP with ECP28MDF (for I atom) level of theory. The lengths of bonds are given in Å.

### 3.2 Cluster Formation Pathways and Free Energy Surface

To explore how MSA affect HIO$_3$-HIO$_2$-based nucleation kinetic, the ACDC simulations were employed to reveal the nucleation mechanism under varying atmospheric conditions. Based on the field measurement (Berresheim et al., 2002; Chen et al., 2018; Sipilä et al., 2016; Beck et al., 2021), the ranges of [MSA], [HIO$_3$], and [HIO$_2$] are set to be $10^6 – 10^8$, $10^6 – 10^8$ and $2.0 \times 10^4 – 2.0 \times 10^6$ molec. cm$^{-3}$, respectively, where [HIO$_3$]/[HIO$_2$] is a constant. Here, the condensation sink (CS) coefficient is set to be $2.0 \times 10^{-3}$ s$^{-1}$ (Dal Maso et al., 2002) and temperature (*T*) is 278 K. Under such conditions, the molecular-

level nucleation pathways and the corresponding branching ratios are depicted in Fig. 3(a). The detailed branch ratio is also shown at 278 K (Fig. S3) and 268 K (Fig. S4). Furthermore, to comprehend how the growth occurs thermodynamically, we herein calculated the Gibbs free energies ($\Delta G$, Eq. (2)) along the main clustering pathway at the conditions of $T = 268 - 278$ K, $[HIO_3] = 1.0 \times 10^6$, $[HIO_2] = 2.0 \times 10^4$, and $[MSA] = 5.0 \times 10^6$ molec. cm$^{-3}$ (Fig. 3(b) and Fig. S5).

As shown in Fig. 3(a), the clustering pathways, at $T = 278$ K, CS $= 2.0 \times 10^{-3}$ s$^{-1}$, $[HIO_3] = 1.0 \times 10^6$, $[HIO_3] = 2.0 \times 10^4$, and $[MSA] = 5.0 \times 10^6$ molec. cm$^{-3}$, can be categorized into two main types: i) MSA-involved pathways, including HIO$_2$-MSA and HIO$_3$-HIO$_2$-MSA nucleation; and ii) non-MSA pathways, primarily involving HIO$_3$-HIO$_2$ nucleation. For the HIO$_2$-MSA pathway, the initial formation of $(HIO_2)_1(MSA)_1$ heterodimer occurs without any energy barrier (Fig. 3(b)). And the subsequent cluster growth mainly proceeds via sequential addition of HIO$_2$ or MSA monomer, partly coupled with cluster collisions. Specifically, 63% of $(HIO_2)_2(MSA)_2$ results from $(HIO_2)_2(MSA)_1$ colliding with MSA monomer with energy barrier of 1.00 kcal mol$^{-1}$, while 36% from barrierless combination of two $(HIO_2)_1(MSA)_1$ cluster. At this point, kinetic drives growth through the collision of $(HIO_2)_2(MSA)_1$ with MSA monomer, instead of following the lowest energy pathway. This is because the collision frequency of $(HIO_2)_1(MSA)_1$ and a HIO$_2$ monomer is relatively higher, stemming from the higher $[HIO_2]$. Then, the formed $(HIO_2)_2(MSA)_2$ further collides with a HIO$_2$ monomer, yielding the stable $(HIO_2)_3(MSA)_2$ cluster against evaporation. As to HIO$_3$-HIO$_2$-MSA nucleation, the formation of $(HIO_3)_1(HIO_2)_3(MSA)_1$ cluster arises from i) the collision of $(HIO_2)_2(MSA)_1$ with $(HIO_3)_1(HIO_2)_1$ (18%, energy barrier: 1.00 kcal mol$^{-1}$), and ii) $(HIO_3)_1(HIO_2)_2$ with $(HIO_2)_1(MSA)_1$ (62%, energy barrier: 0.17 kcal mol$^{-1}$), as well as iii) $(HIO_3)_1(HIO_2)_3$ binding with a MSA monomer (20%, energy barrier: 3.80 kcal mol$^{-1}$). In addition, for the non-MSA pathway marked by purple arrows, 76% of $(HIO_3)_2(HIO_2)_2$ cluster formation arises from the collision between two $(HIO_3)_1(HIO_2)_1$ cluster, which accords closely with the barrierless pathway shown in Fig. 3(b).

Overall, the MSA-involved pathways contribute to 74% of cluster formation, while non-MSA path accounts for only 26%. Although the HIO$_3$-HIO$_2$-MSA growth pathway is less favorable than the HIO$_3$-HIO$_2$ and HIO$_2$-MSA pathways at $T = 278$ K, it can become barrierless at the lower temperature of 268 K (Fig. S5). This result may be explained by the fact that the lower temperature results in a decrease in the evaporation rates of the HIO$_3$-HIO$_2$-MSA clusters. The detailed cluster evaporation paths and corresponding $\gamma$ at 268 K and 278 K are collected in Table S5 and Table S6. Generally, stable clusters have lower evaporation rates. According to the calculated cluster evaporation rates ($\Sigma\gamma$, s$^{-1}$) at 278 K (Table S7), more than 40% of the clusters have $\Sigma\gamma$ less than $10^{-3}$ s$^{-1}$, indicating relatively high stability ($\beta C/\Sigma\gamma > 1$). Among these resulting stable clusters (see Fig. S6), the majority (85%) contains HIO$_2$. Moreover, the concentration of these stable clusters increases gradually with time, even reaching a maximum of $10^4$ molec. cm$^{-3}$ (Fig. S6). Of these stable clusters, initial $(HIO_3)_1(HIO_2)_1$, $(HIO_2)_2$, and $(MSA)_1(HIO_2)_1$ dimer form rapidly, and at t = ~1 s, heterotrimers $(HIO_3)_1(HIO_2)_2$ and $(MSA)_1(HIO_2)_2$ begin to form, after which, the larger-sized clusters also form. These time-dependent evidence suggests that MSA is involved in the whole clustering process, from the initial formation of smaller clusters to the large-sized nucleated clusters that potentially further grow. Taken together, these findings highlight the direct and significant involvement of MSA in HIO$_3$-HIO$_2$ nucleation, facilitating cluster formation.

**(a) Cluster formation pathway**

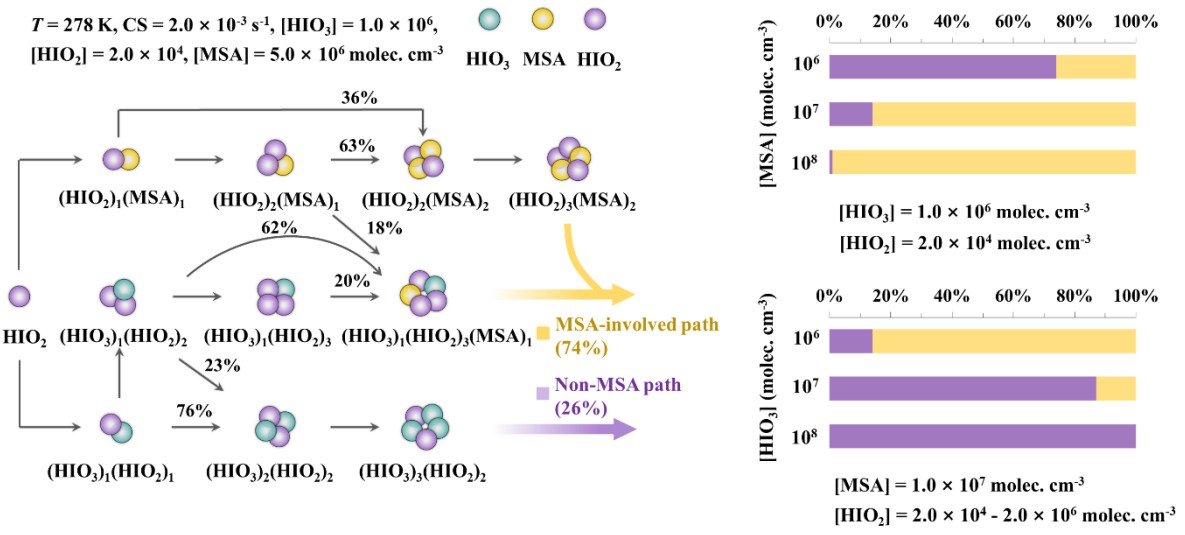

Branch ratio of flux out ($T$ = 278 K, CS = 2.0×10⁻³ s⁻¹)

$T$ = 278 K, CS = 2.0 × 10⁻³ s⁻¹, [HIO₃] = 1.0 × 10⁶, [HIO₂] = 2.0 × 10⁴, [MSA] = 5.0 × 10⁶ molec. cm⁻³

[HIO₃] = 1.0 × 10⁶ molec. cm⁻³
[HIO₂] = 2.0 × 10⁴ molec. cm⁻³

[MSA] = 1.0 × 10⁷ molec. cm⁻³
[HIO₂] = 2.0 × 10⁴ - 2.0 × 10⁶ molec. cm⁻³

**(b) Free energy surface of cluster formation**

$T$ = 278K, [HIO₃] = 1.0 × 10⁶, [HIO₂] = 2.0 × 10⁴, [MSA] = 5.0 × 10⁶ molec. cm⁻³

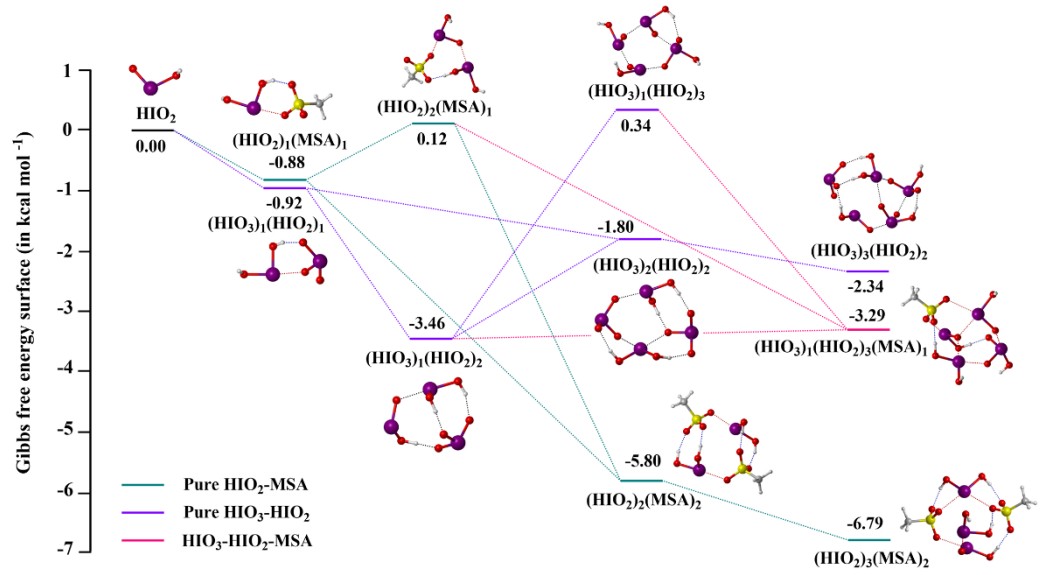

**Figure 3. (a)** Left: main cluster growth pathway of the HIO₃-HIO₂-MSA nucleation system at $T$ = 278K, CS = 2.0 × 10⁻³ s⁻¹, [HIO₃] = 1.0 × 10⁶, [HIO₂] = 2.0 × 10⁴, and [MSA] = 5.0 × 10⁶ molec. cm⁻³. Right: branch ratio of flux out under varying [MSA] (10⁶ –10⁸ molec. cm⁻³) and [HIO₃] (10⁶ – 10⁸ molec. cm⁻³). The yellow and purple arrows (or bar) denote MSA-involved and non-MSA flux out (or branch ratio), respectively. **(b)** The Gibbs free energies of cluster formation (Δ$G$, kcal mol⁻¹) based on the main clustering pathway in HIO₃-HIO₂-MSA nucleation system. [HIO₃]/[HIO₂] is a constant.

185

190

As presented in the right of Figure 3(a), the contribution of different clustering pathways to the flux out varies with precursor concentrations. With increasing [MSA] from $10^6$ to $10^8$ molec. cm$^{-3}$, the contribution of MSA-involved pathways rises from 1% to 99% during nucleation. And at the median [MSA] of $10^7$ molec. cm$^{-3}$, the MSA-involved pathway contributes 86%, whereas the non-MSA pathway accounts for just 14%. In contrast, the ratio of MSA-involved pathways decreases (from 86% to 0%) with increasing concentrations of iodine oxoacids. At higher [HIO$_3$] of $10^8$ molec. cm$^{-3}$, the HIO$_3$-HIO$_2$ pathway dominates nucleation. Predictably, the kinetic impact of MSA on HIO$_3$-HIO$_2$ nucleation is more pronounced in marine regions with richer MSA away from iodine sources.

### 3.3 Enhancement on Cluster Formation Rates

Guided by the clustering pathway analysis, MSA has shown its potential to participate in the HIO$_3$-HIO$_2$-based nucleation, but its detailed impacts on cluster formation rates ($J$, cm$^{-3}$ s$^{-1}$) remain uncertain. Herein, the influence of MSA on $J$ under different atmospheric conditions are systematically analyzed below.

Figure 4 presents the simulated $J$ of HIO$_3$-HIO$_2$-MSA (red bar) and HIO$_3$-HIO$_2$ system (grey bar) against the varying temperatures ($T = 258 – 298$ K) at CS $= 2.0 \times 10^{-3}$ s$^{-1}$, [HIO$_3$] $= 1.0 \times 10^7$, [HIO$_2$] $= 2.0 \times 10^5$, and [MSA] $= 1.0 \times 10^7$ molec. cm$^{-3}$. Clearly, $J$(HIO$_3$-HIO$_2$-MSA) is consistently higher than $J$(HIO$_3$-HIO$_2$), highlighting the enhancement of MSA on HIO$_3$-HIO$_2$-based clustering under common atmospheric temperatures. Specifically, both $J$(HIO$_3$-HIO$_2$-MSA) and $J$(HIO$_3$-HIO$_2$) are negatively dependent on $T$, due to reduced cluster evaporation caused by low $T$. As a result, MSA could promote nucleation with higher $J$, especially at the colder regions, such as polar oceans.

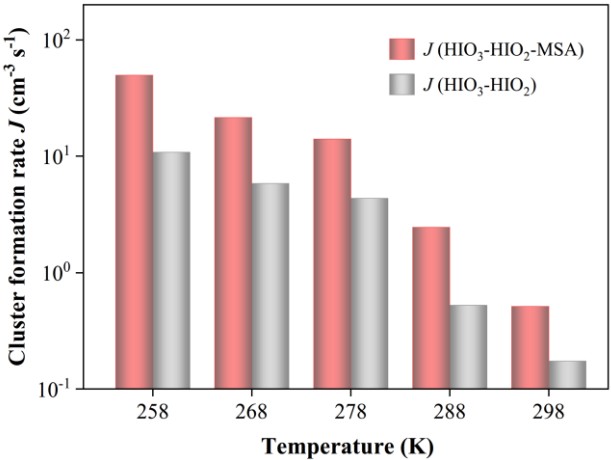

**Figure 4.** Simulated cluster formation rates $J$ (cm$^{-3}$ s$^{-1}$) against varying atmospheric temperatures: $T = 258 – 298$ K, CS $= 2.0 \times 10^{-3}$ s$^{-1}$, [HIO$_3$] $= 1.0 \times 10^7$, [HIO$_2$] $= 2.0 \times 10^5$, and [MSA] $= 1.0 \times 10^7$ molec. cm$^{-3}$.

In fact, apart from atmospheric temperature, precursor concentrations may vary regionally or seasonally, further affecting nucleation. So, to comprehensively reveal the effect of MSA, here we defined and calculated MSA-driven enhancement factor $R$ (Eq. (6)) under varying concentrations of MSA, HIO$_3$, and HIO$_2$ (i.e., [MSA], [HIO$_3$], and [HIO$_2$], unit: molec. cm$^{-3}$).

$$R = \frac{J(\text{HIO}_3\text{-HIO}_2\text{-MSA})}{J(\text{HIO}_3\text{-HIO}_2)} = \frac{J([\text{HIO}_3] = x, [\text{HIO}_2] = y, [\text{MSA}] = z)}{J([\text{HIO}_3] = x, [\text{HIO}_2] = y)}, \tag{6}$$

where $J(\text{HIO}_3\text{-HIO}_2\text{-MSA})$ and $J(\text{HIO}_3\text{-HIO}_2)$ represent the cluster formation rate of $\text{HIO}_3\text{-HIO}_2\text{-MSA}$ and $\text{HIO}_3\text{-HIO}_2$ system, respectively. $x$, $y$ and $z$ are the adopted $[\text{HIO}_3]$, $[\text{HIO}_2]$, and $[\text{MSA}]$, respectively.

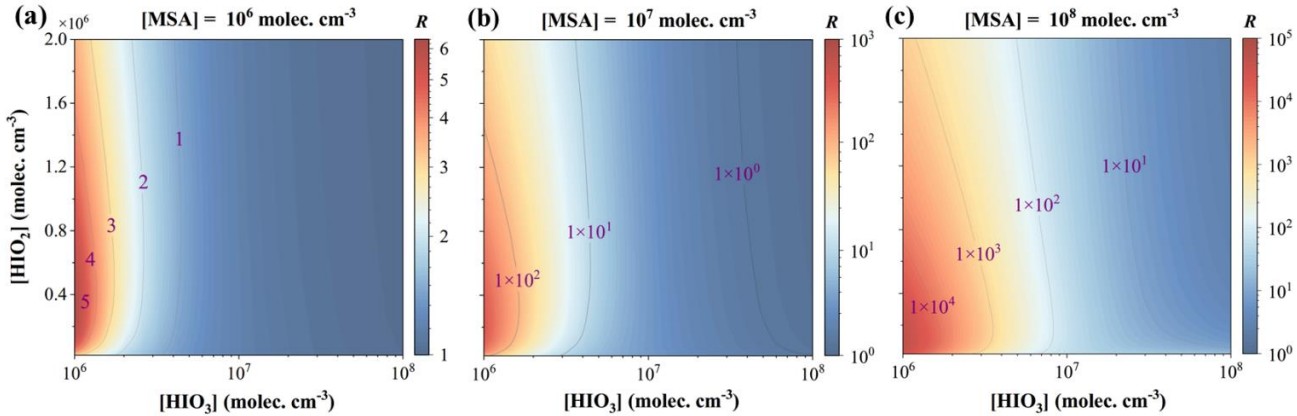

**Figure 5.** Enhancement strength $R$ of MSA on cluster formation rates at varying precursor concentrations: $[\text{HIO}_3] = 10^6 - 10^8$, $[\text{HIO}_2] = 2.0 \times 10^4 - 2.0 \times 10^6$ molec. cm$^{-3}$, **(a)** $[\text{MSA}] = 1.0 \times 10^6$ molec. cm$^{-3}$, **(b)** $[\text{MSA}] = 1.0 \times 10^7$ molec. cm$^{-3}$, and **(c)** $[\text{MSA}] = 1.0 \times 10^8$ molec. cm$^{-3}$, $T = 278$ K, CS $= 2.0 \times 10^{-3}$ s$^{-1}$.

As seen from Fig. 5, with $[\text{MSA}]$ ranging from $10^6$ to $10^8$ molec. cm$^{-3}$ (Fig. 5(a) – (c)), the maximum $R$ increases from 5 to $10^4$-fold, which is primarily due to MSA-mediated synergistic nucleation with $\text{HIO}_3$ and $\text{HIO}_2$ (recalling Sect. 3.2). Even at a median $[\text{MSA}]$ of $1.0 \times 10^7$ molec. cm$^{-3}$, the resulting $R$ can reach approximately $10^2$-fold. In contrast, $R$ is decayed at conditions of higher $[\text{HIO}_3]$ and $[\text{HIO}_2]$. Furthermore, at the conditions with lower $[\text{HIO}_3]/[\text{HIO}_2]$, where $R$ is higher, the contribution of MSA nucleating with $\text{HIO}_2$ increase due to the relative scarcity of $\text{HIO}_3$. Conversely, $R$ decreases at higher $[\text{HIO}_3]/[\text{HIO}_2]$, i.e., the impacts of MSA decreases. That is, the enhancing effect of MSA on $J$ is limited in near-iodine source regions. Naturally, in regions with sparser iodine, the promoting effect of MSA is significant. However, the atmospheric $[\text{HIO}_3]$ ranges widely from $10^6$ to $10^8$ molec. cm$^{-3}$. When $[\text{HIO}_3]$ is comparable or higher than $[\text{MSA}]$, the $\text{HIO}_3\text{-HIO}_2$ pathway contributes more, and the $R$ of MSA decreases with the rising $[\text{HIO}_3]$. It is worth noting that when $[\text{HIO}_3]$ is comparable to $[\text{MSA}]$, the $R$ of MSA is greater than 2, as the contribution of MSA to clustering includes not only the direct formation of $\text{HIO}_3\text{-HIO}_2\text{-MSA}$ clusters (~20%), but also its 'catalysis' role in facilitating formation of initial $\text{HIO}_3\text{-HIO}_2$ clusters (Fig. S7). To sum up, MSA can promote nucleation, particularly in marine regions characterized by lower $T$, lower $[\text{HIO}_3]$ and $[\text{HIO}_2]$. In addition, we also considered the conditions in relatively polluted (CS $= 1.0 \times 10^{-2}$ s$^{-1}$) and clean (CS $= 1.0 \times 10^{-4}$ s$^{-1}$) environment and found that, similar to the environment with CS value of $2.0 \times 10^{-3}$ s$^{-1}$, MSA exhibits significant promoting effect on iodine particle formation (Figs. S8 – S11). Furthermore, the effect of $\text{HIO}_2$ addition on the whole nucleation system was considered, as it is not only the rate-limiting step for cluster formation, leading to the significant increasement of the

$J$(HIO$_3$-HIO$_2$-MSA) compared to $J$(HIO$_3$-MSA) (Fig. S12), but also thermodynamically favorable due to HIO$_3$-HIO$_2$-MSA path is almost barrierless (1.24 kcal mol$^{-1}$) compared to HIO$_3$-MSA pathway (Fig. S13).

### 3.4 Comparison with Field Observations

To further assess the atmospheric implication of the proposed HIO$_3$-HIO$_2$-MSA nucleation, we herein simulated $J$ in Fig. 6 based on the ambient conditions of the typical polar regions (e.g., Ny-Ålesund and Marambio) and the mid-latitude marine regions (e.g., Mace Head and Réunion). Subsequently, we compared these simulation results with observed nucleation rates and the definition of cluster formation rate was detailed in Supporting Information (SI). As shown in Fig. 6(a), the $J$(HIO$_3$-HIO$_2$-MSA) simulated at $T$ = 268 K, CS = $4.0 \times 10^{-4}$ s$^{-1}$, [HIO$_3$] = $10^5 - 10^6$, [HIO$_2$] = $2.0 \times 10^3 - 2.0 \times 10^4$, and [MSA] = $10^6$

$- 10^8$ molec. cm$^{-3}$ was compared with field observations in coastal Ny-Ålesund (Beck et al., 2021; He et al., 2021). Both $J$(HIO$_3$-HIO$_2$-MSA) and $J$(HIO$_3$-HIO$_2$) increase with the rising [HIO$_3$] and [HIO$_2$]. Importantly, the addition of MSA effectively promotes $J$ to a higher level (orange area), aligning with most field measurement ($10^{-3} - 10^{-1}$ cm$^{-3}$ s$^{-1}$, gray lines) (Beck et al., 2021). Even when [MSA] is as low as $1.0 \times 10^6$ molec. cm$^{-3}$ (the orange line below), the $J$(HIO$_3$-HIO$_2$-MSA) can be one order of magnitude higher than the observed $J$ of $10^{-3}$ cm$^{-3}$ s$^{-1}$ (the gray line below). Moreover, the simulated $J$ in Fig.

6(b) was obtained at the conditions of coastal Marambio, Antarctic: $T$ = 273 K, CS = $1.0 \times 10^{-4}$ s$^{-1}$, [HIO$_3$] = $10^5 - 10^6$, [HIO$_2$] = $2.0 \times 10^3 - 2.0 \times 10^4$, and [MSA] = $10^6 - 10^7$ molec. cm$^{-3}$ (Quéléver et al., 2022; He et al., 2021). Compared to $J$(HIO$_3$-HIO$_2$), the MSA-enhanced $J$(HIO$_3$-HIO$_2$-MSA) is overall higher, better fitting with the field observations ($10^{-1} - 2.0 \times 10^1$ cm$^{-3}$ s$^{-1}$, gray area) (Quéléver et al., 2022). These findings imply that MSA potentially plays a vital role in cold polar oceanic regions, particularly with higher [MSA] during NPF events.

In addition, the influence of HIO$_3$-HIO$_2$-MSA nucleation over the relatively warmer mid-latitude marine areas, such as Mace Head and Réunion, was investigated here (Fig. S14). We found that $J$(HIO$_3$-HIO$_2$-MSA) is slightly higher than $J$(HIO$_3$-HIO$_2$), especially at regions with high concentrations of iodine oxoacids (e.g., Mace Head), showing a relatively limited enhancement of MSA on nucleation. Based on the simulated $J$(HIO$_3$-HIO$_2$) (~$10^4$ cm$^{-3}$ s$^{-1}$), iodine nucleation can pretty much explain the NPF events of Mace Head (Fig. S14(a)), which provides potential theoretical evidence for explaining the previous

findings (Sipilä et al., 2016).

Overall, at the mid-latitude oceans, especially near iodine source like Mace Head, MSA may have limited enhancement on nucleation. In this case, abundant iodine oxoacids dominate clustering process. While in the colder polar regions, particularly with higher [MSA] like Marambio, MSA indeed significantly facilitate HIO$_3$-HIO$_2$ nucleation, suggesting a vital role in polar NPF.

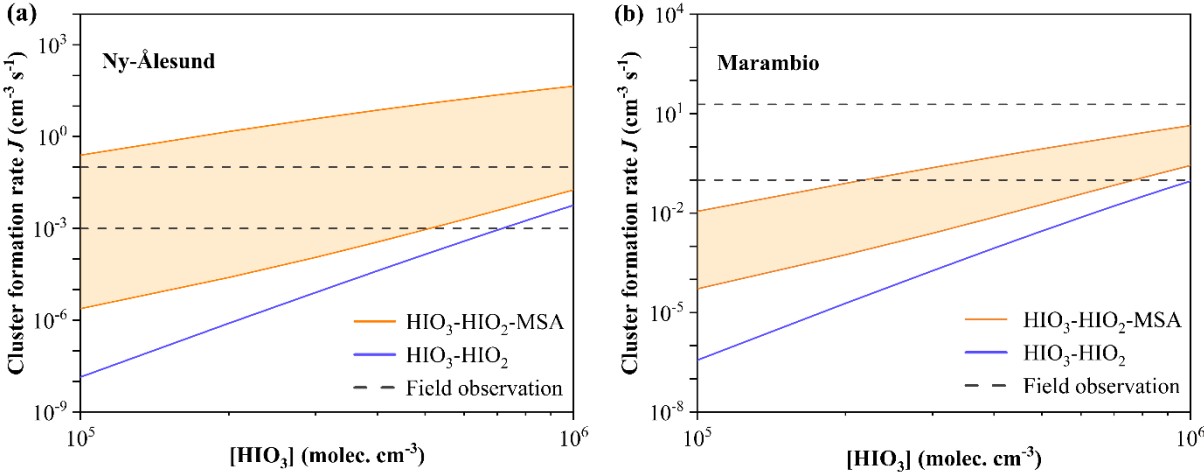

**Figure 6.** Comparison with the simulated cluster formation rates ($J$, cm$^{-3}$ s$^{-1}$) and field observations at the ambient conditions of **(a)** Ny-Ålesund ($T$ = 268 K, CS = 4.0 × 10$^{-4}$ s$^{-1}$, [HIO$_3$] = 10$^5$ – 10$^6$, [HIO$_2$] = 2.0 × 10$^3$ – 2.0 × 10$^4$, and [MSA] = 10$^6$ – 10$^8$ molec. cm$^{-3}$), **(b)** Marambio ($T$ = 273 K, CS = 1.0 × 10$^{-4}$ s$^{-1}$, [HIO$_3$] = 10$^5$ – 10$^6$, [HIO$_2$] = 2.0 × 10$^3$ – 2.0 × 10$^4$, and [MSA] = 10$^6$ – 10$^7$ molec. cm$^{-3}$). The orange area, blue line and gray area represent $J$(HIO$_3$-HIO$_2$-MSA), $J$(HIO$_3$-HIO$_2$), and $J$(Field observation), respectively. [HIO$_3$]/[HIO$_2$] is a constant.

## 4 Conclusion

The present study systematically investigates HIO$_3$-HIO$_2$-based nucleation process enhanced by MSA at the molecular level by QC calculations and ACDC simulations. The results indicate that MSA can structurally stabilize HIO$_3$-HIO$_2$-based clusters by building the intricate networks with more HBs and XBs. Also, during clustering, MSA replaces HIO$_3$ in protonating HIO$_2$ to form ion pairs, resulting in relatively strong electrostatic attractions. Moreover, thermodynamic analyses suggest that MSA-involved clustering is nearly barrierless. Compared to previously reported HIO$_3$-HIO$_2$ system, the MSA-involved synergistic nucleation with HIO$_3$ and HIO$_2$ proceeds more efficiently, through two additional clustering pathways: i) HIO$_2$-MSA binary and ii) HIO$_3$-HIO$_2$-MSA ternary pathway. And the resulting enhancement of MSA on nucleation is stronger at colder regions, especially with richer MSA, but weaker in the environments near iodine source. Further comparison with field observations indicates that the HIO$_3$-HIO$_2$-MSA synergistic nucleation plays a limited role in the mid-latitude oceans, particularly with abundant iodine (e.g., Mace Head), but an important role in the colder polar regions (e.g., Ny-Ålesund and Marambio).

This study highlights the essential enhancing role of MSA in iodine oxoacids nucleation, and the proposed HIO$_3$-HIO$_2$-MSA synergistic nucleation may help to explain the observed abundant iodine particles during marine NPF events. In addition to MSA, given the complex oceanic atmosphere, other potential nucleation precursors, such as sulfuric acid and amines, may

also affect the HIO$_3$-HIO$_2$ nucleation process, further contributing to the formation of marine iodine particles, which deserves future studies.

**Data availability.** The data in this article are available from the corresponding author upon request (anning@bit.edu.cn and zhangxiuhui@bit.edu.cn).

**Supplement.** The supplement related to this article is available online at :

**Author contribution.** XZ designed and supervised the research. JL and NW performed the quantum chemical calculations and the ACDC simulations. JL, NW and AN analyzed data. JL, AN and XZ wrote the paper with contributions from all of the other co-authors.

**Competing interests.** The contact author has declared that neither they nor their co-authors have any competing interests.

**Disclaimer.** Publisher's note: Copernicus Publications remains neutral with regard to jurisdictional claims in published maps
and institutional affiliations.

**Acknowledgement.** We acknowledge the National Supercomputing Center in Shenzhen for providing the computational resources and the Turbomole program.

**Financial support.** This work is supported by the National Science Fund for Distinguished Young Scholars (grant no. 22225607) and the National Natural Science Foundation of China (grant no. 21976015, and no. 22306011). An Ning was also
supported by the China Postdoctoral Science Foundation (grant no. 2023M730236).

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
