# Peer review of "Molecular-level study on the role of methanesulfonic acid in iodine oxoacids nucleation"

_EGUsphere, 2023_

## Author Comment (AC1)

**Responses to Referee #3's comments**

We are grateful to the reviewers for their valuable and helpful comments on our manuscript "**Molecular-level study on the role of methanesulfonic acid in iodine oxoacids nucleation**" (MS No.: egusphere-2023-2084). We have revised the manuscript carefully according to reviewers' comments. The point-to-point responses to the Referee #3's comments are summarized below:

**Referee comments:**

This manuscript investigates the enhancement effects of methanesulfonic acid (MSA) on the iodic acid ($HIO_3$)-iodous acid ($HIO_2$) nucleation system, which has been reported as an important mechanism of marine new particle formation (NPF). The intermolecular interactions, cluster stability, the formation pathway/ free energy surface of cluster formation as well as the enhancement of formation rate of the $HIO_3$-$HIO_2$-MSA ternary nucleation system was systematically studied with the combination of quantum chemical simulation and ACDC approach. This paper provided theoretical evidence that the involvement of MSA can structurally stabilize $HIO_3$-$HIO_2$-based clusters and has positive synergistic effect on the nucleation with $HIO_3$ and $HIO_2$. This manuscript is nicely written and fits the scope of ACP. I recommend the manuscript to be published after the following comments are addressed.

**Response:** We sincerely thank for the reviewer's careful review of our manuscript, as well as the valuable and positive comments.
* * *
**General comments:**

**Comment 1:** Although sufficient theoretical details of the cluster conformations have been provided, the authors should also clarify the definition of "stable cluster". We should derive a cluster's concentration by the competition between its collisional formation and evaporation, instead of only judging the formation free energy. Since the authors have conducted ACDC calculations, the evaporation rates of the major clusters should be discussed, as well as the time dependent concentration variations of these major clusters.

**Response:** Thanks for the reviewer's professional suggestions. Accordingly, we have added the definition of "stable cluster", which is determined based on the competition between cluster

collisional formation and evaporation. A cluster is deemed stable when its collisional formation dominates over evaporation[1]. And the corresponding definition has been added in the revised manuscript (Lines 102-103, page 4) as follows: "Additionally, whether the clusters in the simulated system are stable depends on whether the rate of collision frequencies exceeds the total evaporation rate coefficients ($\beta C/\Sigma\gamma > 1$) (Table S4)."

According to the helpful suggestion of the reviewer, the evaporation rates of the major clusters have been discussed in Lines 175-178 of page 7 as follows: "Generally, stable clusters have lower evaporation rates. According to the calculated cluster evaporation rates ($\Sigma\gamma$, s$^{-1}$) at 278 K (Table S7), more than 40% of the clusters have $\Sigma\gamma$ less than $10^{-3}$ s$^{-1}$, indicating relatively high stability ($\beta C/\Sigma\gamma > 1$). Among these resulting stable clusters (see Fig. S6), the majority (85%) contains HIO$_2$."

[Figure]

**Figure S6.** The concentration (molec. cm$^{-3}$) of stable clusters in HIO$_3$-HIO$_2$-MSA system as a function of time, at $T$ = 278 K, CS = $2.0 \times 10^{-3}$ s$^{-1}$, [HIO$_3$] = $10^6$, [HIO$_2$] = $2.0 \times 10^4$, [MSA] = $5.0 \times 10^6$ molec. cm$^{-3}$.

Further, as suggested by the reviewer, we calculated the time-dependent concentration variations of the mentioned major cluster sites and presented the simulation results in Fig. S6. The corresponding analysis has been included in the revised manuscript (Line 178-183, page 7), and a copy is provided below: "Moreover, the concentration of these stable clusters increases gradually with time, even reaching a maximum of $10^4$ molec. cm$^{-3}$ (Fig. S6). Of these stable clusters, initial (HIO$_3$)$_1$(HIO$_2$)$_1$, (HIO$_2$)$_2$, and (MSA)$_1$(HIO$_2$)$_1$ dimer form rapidly, and at t = ~1

s, heterotrimers $(HIO_3)_1(HIO_2)_2$ and $(MSA)_1(HIO_2)_2$ begin to form, after which, the larger-sized clusters also form. These time-dependent evidence suggests that MSA is involved in the whole clustering process, from the initial formation of smaller clusters to the large-sized nucleated clusters that potentially further grow."
* * *
**Comment 2:** I find the results in section 3.2 a bit confusing when trying to interpret the relative importance of the MSA-involved path in $HIO_3$-$HIO_2$-MSA nucleation under different conditions. For example, Figure 3a (upper panel) shows that the MSA-involved path contributes to 74 % of the cluster formation, under [MSA] of $5.0 \times 10^6$ molec. cm$^{-3}$, [HIO$_3$] of $1.0 \times 10^6$ molec. cm$^{-3}$, while the MSA-involved path contributes to ~20 % under [MSA] of $1.0 \times 10^6$ molec. cm$^{-3}$, [HIO$_3$] of $1.0 \times 10^6$ molec. cm$^{-3}$. This result indicates that MSA is less efficient in clustering with HIO$_2$, comparing with HIO$_3$. The authors have mentioned the atmospheric concentration level of MSA in line 40, the authors should also include the discussion about the concentration of iodine oxoacids in the revised manuscript. Since if the concentration of HIO$_3$ is comparable or higher than MSA, the scenario of ACDC simulation cannot reflect the condition of real atmosphere.

**Response:** Thanks for the reviewer's insightful comments. As analyzed by the reviewer, MSA exhibits lower efficiency in clustering with HIO$_2$ compared to HIO$_3$. Thus, we agree with the reviewer's suggestion to discuss the conditions with varying iodine oxoacids concentration, as it is very necessary. According to the scenarios presented in Fig. 3(a) (right panel) and Fig. 5 where [HIO$_3$] is comparable or higher than [MSA], the corresponding analyses were supplemented in the revised manuscript (Line 227-231) and copied below: "However, the atmospheric [HIO$_3$] ranges widely from $10^6$ to $10^8$ molec. cm$^{-3}$. When [HIO$_3$] is comparable or higher than [MSA], the HIO$_3$-HIO$_2$ pathway contributes more, and the $R$ of MSA decreases with the rising [HIO$_3$]. It is worth noting that when [HIO$_3$] is comparable to [MSA], the $R$ of MSA is greater than 2, as the contribution of MSA to clustering includes not only the direct formation of HIO$_3$-HIO$_2$-MSA clusters (~20%), but also its 'catalysis' role in facilitating formation of initial HIO$_3$-HIO$_2$ clusters (Fig. S7)."
* * *
**Comment 3:** The comparison with field measurements seems to be a bit arbitrary. What's the atmospheric condition such as precursor concentration and temperature of the measurement sites reported? What's the definition of the $J$ in the reported field measurements? Can the reported $J$ be directly compared with the ACDC simulated $J$? Moreover, in Line 222, "the observed $J$ of $2.1 \times 10^{-4}$ cm$^{-3}$ s$^{-1}$". This $J$ value is too low for a typical NPF event. Is this value obtained from a non-NPF day? Please check the original paper. More explanation and discussion are needed in this section, which can sharpen the significance of the theoretical study on the merit of atmospheric implication.

**Response:** Thanks for the reviewer's professional comments.

**Item 1) from the reviewer:** The comparison with field measurements seems to be a bit arbitrary. What's the atmospheric condition such as precursor concentration and temperature of the measurement sites reported?

**Response: (a) Ny-Ålesund:** According to Beck[2] et al., Ny-Ålesund is surrounded by open waters throughout the whole year, with the average annual temperature of -5°C [3]. As shown in Figure 1(b) of Beck et al., the ranges of [HIO$_3$] and [MSA] are $10^5 – 10^6$ and $10^6 – 10^8$ molec. cm$^{-3}$, respectively. And the author mentioned that "An explanation for this could be the very low condensation sink of $\sim 4 \times 10^{-4}$ s$^{-1}$ at Ny-Ålesund…".

Thus, the simulation conditions were set to: $T = 268$ K, CS $= 4.0 \times 10^{-4}$ s$^{-1}$, [HIO$_3$] $= 10^5 – 10^6$, [HIO$_2$] $= 2.0 \times 10^3 – 2.0 \times 10^4$, and [MSA] $= 10^6 – 10^8$ molec. cm$^{-3}$.

**(b) Marambio:** According to the description of Marambio[4], many sunny days are observed, occurring with ambient temperatures above 0 °C. The author mentioned that the measured gas-phase concentrations of the species of interest showed maxima of $\sim 2.3 \times 10^7$, and $\sim 3.6 \times 10^6$ molecules cm$^{-3}$ for the total MSA and HIO$_3$ concentration, respectively.

Combined with the ranges of [HIO$_3$] and [MSA] from Figures 5 and 6 in the original paper describing Marambio[4], thus, the simulation conditions were set to: $T = 273$ K, CS $= 1.0 \times 10^{-4}$ s$^{-1}$, [HIO$_3$] $= 10^5 – 10^6$, [HIO$_2$] $= 2.0 \times 10^3 – 2.0 \times 10^4$, and [MSA] $= 10^6 – 10^7$ molec. cm$^{-3}$).

**(c) Mace Head:** In Mace Head, the concentration of HIO$_3$ during the new particle formation events reached $10^8$ molecules cm$^{-3}$, and the range of [HIO$_3$] is set to $10^6 – 10^8$ molec. cm$^{-3}$ according to Figure 1(b) [5]. Moreover, Berresheim et al. reported that the range of [MSA] is $10^5 – 10^7$ molec. cm$^{-3}$, and the temperature can reach to 14°C, as shown in Figure 6 [6].

Thus, the simulation conditions were set to: $T = 287$ K, CS = $2.0 \times 10^{-3}$ s$^{-1}$, [HIO$_3$] = $10^7$ – $10^8$, [HIO$_2$] = $2.0 \times 10^5$ – $2.0 \times 10^6$, and [MSA] = $10^6$ – $10^7$ molec. cm$^{-3}$.

**(d) Réunion:** According to Salignat et al., the average [HIO$_3$] is $2.90 \times 10^5$ molec. cm$^{-3}$, and the ranges of [HIO$_3$] and [MSA] are $10^5$ – $10^7$ and $10^6$ – $10^7$ molec. cm$^{-3}$, respectively, according to Figure 8(a) in the original paper about Réunion [7]. As shown in Figure 4(a), the temperature ranges from 10 to 20 °C.

Thus, the simulation conditions were set to: $T = 288$ K, CS = $2.0 \times 10^{-3}$ s$^{-1}$, [HIO$_3$] = $10^5$ – $3.0 \times 10^6$, [HIO$_2$] = $2.0 \times 10^3$ – $6.0 \times 10^4$, and [MSA] = $10^6$ – $10^8$ molec. cm$^{-3}$.

**Item 2) from the reviewer:** What's the definition of the $J$ in the reported field measurements? Can the reported $J$ be directly compared with the ACDC simulated $J$?

**Response:** In the ACDC simulation, nucleation generally refers to the formation of relatively stable clusters for which collisions with molecules can be assumed to dominate over cluster evaporation. Accordingly, the cluster formation rate ($J$) indicates the particle flux out of the studied system. In this case, it is the rate of clusters forming at some specific size (*i.e.* the net flux into the size from all other sizes)[1]. In field observation, the formation rates ($J_{1.5}$) were measured by instruments, such as nitrate chemical ionization atmospheric pressure interface Time-Of-Flight mass spectrometer (CI-APi-TOF)[2], differential mobility particle sizer (DMPS) and neutral cluster and air ion spectrometer (NAIS)[4].

According to the Kerminen-Kulmala equation[8], cluster formation rates for $d_2$ nm clusters ($J_{d_2}$) relate to those for $d_1$ nm clusters ($J_{d_1}$) by

$$J_{d_1} = J_{d_2} \exp \left\{ \gamma \left( \frac{1}{d_1} - \frac{1}{d_2} \right) \frac{CS}{GR_{d_2-d_1}} \right\}$$

where the $GR_{d_2-d_1}$ is the initial cluster growth rate from $d_1$ to $d_2$ nm, and CS represents condensation sink of clusters by preexisting particles. The parameter $\gamma$ depends on many factors but can usually be approximated by assuming it to be equal to 0.23 nm$^2$ m$^2$ h$^{-1}$.

In this study, the relationship between the formation rates of simulated clusters ($J_{1.2}$) and that of observed clusters ($J_{1.5}$) can be written as:

$$J_{1.2} = J_{1.5} \exp \left\{ 0.23 \times \left( \frac{1}{1.2} - \frac{1}{1.5} \right) \frac{CS}{GR} \right\}$$

where GR was measured to be 3.2 – 4.4 nm·h$^{-1}$ in the 1.1 – 2.0 nm size range during three observed events [9, 10], and CS was 0.002 s$^{-1}$. $J_{1.2}$ was then calculated to be 1.00001 – 1.00002

times of $J_{1.5}$. Thus, the observed cluster formation rates for 1.5 nm clusters can be directly comparable with the simulated $J_{1.2}$.

We have included corresponding justification in Section 3.4 of the revised manuscript (Lines 242-243, Page 11) and supporting file as follows: "Subsequently, we compared these simulation results with observed nucleation rates and the definition of cluster formation rate was detailed in Supporting Information (SI)."

**Item 3) from the reviewer:** Moreover, in Line 222, "the observed $J$ of $2.1 \times 10^{-4}$ cm$^{-3}$ s$^{-1}$". This $J$ value is too low for a typical NPF event. Is this value obtained from a non-NPF day? Please check the original paper.

**Response:** Following the reviewer's suggestion, we have carefully checked the original paper. According to the captions of Fig. 2 ("Examples representing seasonal behavior of NPF observed at Villum and Ny-Ålesund")[2], the present $J$ of $2.1 \times 10^{-4} – 10^{-1}$ cm$^{-3}$ s$^{-1}$ (Figure 2 (c5)) originate from the observed NPF data values during May 4, 2017. Notably, as professionally judged by the reviewer, Beck et al. have also clarified that data with $J$-values < a few $10^{-3}$ cm$^{-3}$ s$^{-1}$ (including the mentioned lower $J$ of $2.1 \times 10^{-4}$ cm$^{-3}$ s$^{-1}$) are highly unreliable and reflect mainly the noise levels[2]. And thus, these low and uncertain $J$ values hardly correspond to NPF events.

Therefore, we have adjusted the range of the observed $J$ of Ny-Ålesund to a reliable range of $10^{-3} – 10^{-1}$ cm$^{-3}$ s$^{-1}$ in Fig. 6(a). Meanwhile, the related statement "…the $J$(HIO$_3$-HIO$_2$-MSA) can be two orders of magnitude higher than the observed $J$ of $2.1 \times 10^{-4}$ cm$^{-3}$ s$^{-1}$" has been changed to "…the $J$(HIO$_3$-HIO$_2$-MSA) can be one order of magnitude higher than the observed $J$ of $10^{-3}$ cm$^{-3}$ s$^{-1}$" in Lines 248-249 (page 11) of the revised manuscript.
* * *
**Specific comments:**

**Comment 4:** It would be preferable to avoid including reference in the abstract.

**Response:** The reference in the abstract has been removed in the revised manuscript.
* * *
**Comment 5:** Abbreviations such as ESP HB XB should be explained at least once in the main text.

**Response:** Thanks for the reviewer's careful reading, all the abbreviations, such as ESP (Electrostatic potential), HB (Hydrogen bond) and XB (Halogen bond), have been explained in their first appearance in the revised manuscript.
* * *
**Comment 6:** Line 47, " Furthermore, given the coexistence of MSA and $HIO_3$ in different marine regions (Quéléver et al., 2022; Beck et al., 2021), the $HIO_3$-$HIO_2$-MSA nucleating mechanism may differ under distinct ambient conditions, but it remains unrevealed." First, nucleating should be nucleation. Second, I feel a bit confusion about this sentence. Is the different $HIO_3$/MSA and $HIO_2$/MSA concentration ratio leads to different nucleation mechanism? The authors have concluded that MSA can promote nucleation, particularly in marine regions characterized by lower T, lower [$HIO_3$] and [$HIO_2$]. It will be more preferable to add some discussion about the [$HIO_3$]/ [$HIO_2$] in different marine atmosphere.

**Response:** Thanks for the reviewer's helpful comments.

**Item 1) from the reviewer:** Line 47, " Furthermore, given the coexistence of MSA and $HIO_3$ in different marine regions (Quéléver et al., 2022; Beck et al., 2021), the $HIO_3$-$HIO_2$-MSA nucleating mechanism may differ under distinct ambient conditions, but it remains unrevealed." First, nucleating should be nucleation.

**Response:** As suggested by the reviewer, all 'nucleating' have been changed to 'nucleation' in the revised manuscript.

**Item 2) from the reviewer:** Second, I feel a bit confusion about this sentence. Is the different $HIO_3$/MSA and $HIO_2$/MSA concentration ratio leads to different nucleation mechanism?

**Response:** As to the mentioned sentence, in fact, here we would like to express that the dominant nucleation mechanism varies with the region. Accordingly, we have rephrased the sentence (Lines 46-48) as follows: "Furthermore, given the coexistence of MSA and $HIO_3$ in different marine regions (Quéléver et al., 2022; Beck et al., 2021), along with the consistent presence of $HIO_3$ and $HIO_2$ as homologous substances[5], the importance of the $HIO_3$-$HIO_2$-MSA nucleation mechanism may differ under distinct ambient conditions, due to their uneven distribution, but it remains unrevealed."

**Item 3) from the reviewer:** The authors have concluded that MSA can promote nucleation, particularly in marine regions characterized by lower T, lower [HIO$_3$] and [HIO$_2$]. It will be more preferable to add some discussion about the [HIO$_3$]/[HIO$_2$] in different marine atmosphere.

**Response:** Furthermore, according to the reviewer's suggestion, we have added the discussion about [HIO$_3$]/[HIO$_2$] in the revised manuscript (Lines 224-226, page 10) as follows: "Furthermore, at the conditions with lower [HIO$_3$]/[HIO$_2$], where *R* is higher, the contribution of MSA nucleation with HIO$_2$ increase due to the relative scarcity of HIO$_3$. Conversely, *R* decreases at higher [HIO$_3$]/[HIO$_2$], i.e., the impacts of MSA decreases."
* * *
**Comment 7:** More explanation of ACDC model setting is needed. For example, the setting of condensation sink and other model parameters.

**Response:** Thanks for reviewer's valuable suggestions. The model parameters in the performed ACDC simulations include: condensation sink (CS), temperature (*T*), precursor concentrations, boundary clusters, collision/evaporation processes (monomer-monomer, monomer-cluster, cluster-cluster). Concerning all the ACDC simulation data presentation, the settings of *T*, CS, and precursor concentrations were provided in the main text and figure captions. And the details on boundary clusters and the employed collision/evaporation processes were added in the Tables S3, S5, and S6 of the Supplementary Information.
* * *
**Comment 8:** Figure 6. The HIO$_3$-HIO$_2$ curve should be an area as the other two.

**Response:** Thanks. The reviewer's comment is important for the clear presentation of the data. Here, we present the rate of HIO$_3$-HIO$_2$ nucleation as a line in Fig. 6, due to the setting of the fixed [HIO$_3$]/[HIO$_2$] in the simulation, given the homology of HIO$_3$ and HIO$_2$, as well as their reported concentration ratio[5]. To avoid potential confusion for readers, we clarified the association between [HIO$_3$] and [HIO$_2$] in the caption of Fig. 6, stating: "[HIO$_3$]/[HIO$_2$] is a constant".
* * *
**Comment 9:** It would be preferable to include some uncertainty analysis.

**Response:** Thanks, this is an important point to improve the robustness of the results. The potential uncertainties may arise from ACDC simulations and quantum chemical (QC)

calculations, thereby we examined how variable ACDC settings, such as condensation sink coefficient (CS), sticking factor (SF, corresponding to a sticking probability for cluster/monomer collision) and the change of calculated Gibbs free energy of cluster formation ($\Delta G$, from quantum chemical calculations) impact the enhancement of MSA ($R_{MSA}$) on cluster formation rate ($J$). Here, the CS values ranged from $1.0 \times 10^{-4}$ s$^{-1}$ to $1.0 \times 10^{-2}$ s$^{-1}$, covering possible CS in relatively clean and polluted regions[10, 11]. The range of SF was set from 0.1 to 1.0 since sticking probabilities for neutral-neutral collisions between 0.1 and 1.0[12].

As shown in the Figs. S15 (a) and (b), although both CS and SF affect $R_{MSA}$ to some extent, the uncertainty range are relatively limited (CS < 32.5% and SF < 17.1%) and the results does not affect the trend and main conclusions.

[Figure]

**Figure S15.** Variation of enhancement strength $R$ of MSA with **(a)** condensation sink coefficient (CS) and **(b)** sticking factor (SF) for HIO$_3$-HIO$_2$-MSA system at $T = 278$ K, [HIO$_3$] $= 1.0 \times 10^7$, [HIO$_2$] $= 2.0 \times 10^5$, and [MSA] $= 1.0 \times 10^7$ molec. cm$^{-3}$.

In addition, the potential uncertainty of quantum chemical calculations is ultimately manifested in the calculated $\Delta G$ values. As reported by Kupiainen[13] et al. (2012), the differences between the computational (DFT//RI-CC2 method) and experimental $\Delta G$ values are about 1 kcal mol$^{-1}$ or less[14]. Accordingly, Almedia[12] et al. (2013) calculated the uncertainty range of ACDC simulated cluster formation resulting from QC calculations by adjusting the binding energy ($\pm 1$ kcal mol$^{-1}$). Further given the consistency of our research framework (DFT//RI-CC2 + ACDC) with Almedia et al. (2013), herein we have performed the uncertainty analysis of $R_{MSA}$ caused by QC calculations through adding or subtracting 1 kcal mol$^{-1}$ from the $\Delta G$ (using $\Delta G_{278K}$ as a reference). The figure below presents the uncertainty analysis results of

$J$ and $R_{MSA}$ at $T = 278$ K, CS = $2.0 \times 10^{-3}$ s$^{-1}$, [HIO$_3$] = $10^7$, [HIO$_2$] = $2.0 \times 10^5$, [MSA] = $10^6$ – $10^8$ molec. cm$^{-3}$.

[Figure]

**Figure S16.** Cluster formation rate $J$ **(a)** and enhancement strength $R$ of MSA **(b)** as a function of [MSA] = $10^6$ – $10^8$ molec. cm$^{-3}$, with different energy of $\Delta G_{278K}$ (black line), $\Delta G_{278K} + 1$ (blue line), $\Delta G_{278K} - 1$ (red line), at $T = 278$ K, CS = $2.0 \times 10^{-3}$ s$^{-1}$, [HIO$_3$] = $10^7$, [HIO$_2$] = $2.0 \times 10^5$ molec. cm$^{-3}$.

Here, we have added the results of $R_{MSA}$ under different CS, SF and $\Delta G$ to the revised supporting file, and for the convenience of the review, we have copied Figures S15-S16 and the corresponding analysis as following: "Here, the potential uncertainties may stem from ACDC simulations and quantum chemical (QC) calculations, we examined the effect of condensation sink coefficient (CS), sticking factor (SF) and calculated $\Delta G$ of clusters on enhancement of MSA to the cluster formation rate. The CS values ranged from $1.0 \times 10^{-4}$ s$^{-1}$ to $1.0 \times 10^{-2}$ s$^{-1}$, covering possible CS in relatively clean and polluted regions[10, 11]. The range of SF was set from 0.1 to 1.0 since sticking probabilities for neutral-neutral collisions between 0.1 and 1.0[12]. Both the CS and SF slightly affect the enhancement of MSA, with limited uncertainty range of CS < 32.5% and SF < 17.1% (Fig. S15). As reported by Kupiainen[13] et al. (2012), the differences between the computational (DFT//RI-CC2 method) and experimental $\Delta G$ values are about 1 kcal mol$^{-1}$ or less[14]. Accordingly, Almedia[12] et al. (2013) calculated the uncertainty range of ACDC simulated cluster formation resulting from QC calculations by adjusting the binding energy ($\pm 1$ kcal mol$^{-1}$). Further given the consistency of our research framework (DFT//RI-CC2 + ACDC) with Almedia et al. (2013), herein we have performed the

uncertainty analysis of $R_{MSA}$ caused by QC calculations through adding or subtracting 1 kcal mol$^{-1}$ from the $\Delta G$ (using $\Delta G_{278K}$ as a reference). As shown in Table S8 and Fig. S16, adjusting the $\Delta G_{278K}$ of clusters by $\pm 1$ kcal mol$^{-1}$ resulted in a minor variation in $J$ and $R$ of MSA, with the overall trend remaining consistent."

**Reference:**

[1] K.-M. Oona, O. Tinja, Atmospheric Cluster Dynamics Code Technical manual, https://github.com/tolenius/ACDC/blob/main/ACDC_Manual_2020_11_25.pdf (2020).

[2] L.J. Beck, N. Sarnela, H. Junninen, C.J.M. Hoppe, O. Garmash, F. Bianchi, M. Riva, C. Rose, O. Peräkylä, D. Wimmer, O. Kausiala, T. Jokinen, L. Ahonen, J. Mikkilä, J. Hakala, X.C. He, J. Kontkanen, K.K.E. Wolf, D. Cappelletti, M. Mazzola, R. Traversi, C. Petroselli, A.P. Viola, V. Vitale, R. Lange, A. Massling, J.K. Nøjgaard, R. Krejci, L. Karlsson, P. Zieger, S. Jang, K. Lee, V. Vakkari, J. Lampilahti, R.C. Thakur, K. Leino, J. Kangasluoma, E.M. Duplissy, E. Siivola, M. Marbouti, Y.J. Tham, A. Saiz-Lopez, T. Petäjä, M. Ehn, D.R. Worsnop, H. Skov, M. Kulmala, V.M. Kerminen, M. Sipilä, Differing Mechanisms of New Particle Formation at Two Arctic Sites, Geophys. Res. Lett. 48 (2021).

[3] NASA POWER. Data access viewer. https://power.larc.nasa.gov/data-accessviewer/, (2022).

[4] L.L.J. Quéléver, L. Dada, E. Asmi, J. Lampilahti, T. Chan, J.E. Ferrara, G.E. Copes, G. Pérez-Fogwill, L. Barreira, M. Aurela, D.R. Worsnop, T. Jokinen, M. Sipilä, Investigation of new particle formation mechanisms and aerosol processes at Marambio Station, Antarctic Peninsula, Atmos. Chem. Phys. 22 (2022) 8417–8437.

[5] M. Sipilä, N. Sarnela, T. Jokinen, H. Henschel, H. Junninen, J. Kontkanen, S. Richters, J. Kangasluoma, A. Franchin, O. peräkylä, M.P. Rissanen, M. Ehn, H. Vehkamäki, T. Kurten, T. Berndt, T. Petäjä, D. Worsnop, D. Ceburnis, V.M. Kerminen, M. Kulmala, C. O'Dowd, Molecular-scale evidence of aerosol particle formation via sequential addition of $HIO_3$, Nature 537 (2016) 532–534.

[6] H. Berresheim, T. Elste, K. Rosman, M. Dal Maso, H.G. Tremmel, J.M. Mäkelä, A.G. Allen, M. Kulmala, H.-C. Hansson, Gas-aerosol relationships of $H_2SO_4$, MSA, and OH: Observations in the coastal marine boundary layer at Mace Head, Ireland, J. Geophys. Res.-Atmos. 107 (2002) PAR 5-1–PAR 5-12.

[7] R. Salignat, M. Rissanen, S. Iyer, J.-L. Baray, P. Tulet, J.-M. Metzger, J. Brioude, K. Sellegri, C. Rose, Measurement Report: Insights into the chemical composition of molecular clusters present in the free troposphere over the Southern Indian Ocean: observations from the Maïdo observatory (2150 m a.s.l., Reunion Island), EGUsphere (2023) 1–41.

[8] M. Kulmala, T. Petäjä, T. Nieminen, M. Sipilä, H.E. Manninen, K. Lehtipalo, M. Dal Maso, P.P. Aalto, H. Junninen, P. Paasonen, I. Riipinen, K.E.J. Lehtinen, A. Laaksonen, V.-M. Kerminen, Measurement of the nucleation of atmospheric aerosol particles, Nature Protocols 7 (2012) 1651-1667.

[9] D. Xia, J. Chen, H. Yu, H.B. Xie, Y. Wang, Z. Wang, T. Xu, D.T. Allen, Formation Mechanisms of Iodine-Ammonia Clusters in Polluted Coastal Areas Unveiled by Thermodynamics and Kinetic Simulations, Environ. Sci. Technol. 54 (2020) 9235-9242.

[10] H. Yu, L. Ren, X. Huang, M. Xie, J. He, H. Xiao, Iodine speciation and size distribution in ambient aerosols at a coastal new particle formation hotspot in China, Atmos. Chem. Phys. 19 (2019) 4025–4039.

[11] X.-C. He, Y.J. Tham, L. Dada, M. Wang, H. Finkenzeller, D. Stolzenburg, S. Iyer, M. Simon, A.K. ¨rten, J. Shen, B. Roerup, M. Rissanen, S. Schobesberger, R. Baalbaki, D.S. Wang, T.K. Koenig, T. Jokinen, N. Sarnela, L.J. Beck, J. Almeida, S. Amanatidis, A. Amorim, F. Ataei, A. Baccarini, B. Bertozzi, F. Bianchi, S. Brilke, L. Caudillo, D. Chen, R. Chiu, B. Chu, A. Dias, A. Ding, J. Dommen, J. Duplissy, I.E. Haddad, L.G. Carracedo, M. Granzin, A. Hansel, M. Heinritzi, V. Hofbauer, H. Junninen, J. Kangasluoma, D. Kemppainen, C. Kim, W. Kong, J.E. Krechmer, A. Kvashin, T. Laitinen, H. Lamkaddam, C.P. Lee, K. Lehtipalo, M. Leiminger, Z. Li, V. Makhmutov, H.E. Manninen, G. Marie, R. Marten, S. Mathot, R.L. Mauldin, B. Mentler, O. Moehler, T. Mueller, W. Nie, A. Onnela, T. Petaja, J. Pfeifer, M. Philippov, A. Ranjithkumar, A. Saiz-Lopez, I. Salma, W. Scholz, S. Schuchmann, B. Schulze, G. Steiner, Y. Stozhkov, C. Tauber, A. Tome, R.C. Thakur, O. Vaisanen, M. Vazquez-Pufleau, A.C.

Wagner, Y. Wang, S.K. Weber, P.M. Winkler, Y. Wu, M. Xiao, C. Yan, Q. Ye, A. Ylisirnio, M. Zauner-Wieczorek, Q. Zha, P. Zhou, R.C. Flagan, J. Curtius, U. Baltensperger, M. Kulmala, V.-M. Kerminen, T. Kurten, N.M. Donahue, R. Volkamer, J. Kirkby, D.R. Worsnop, M. Sipila, Role of iodine oxoacids in atmospheric aerosol nucleation, Science 371 (2021) 589–595.

[12] J. Almeida, S. Schobesberger, A. Kürten, I.K. Ortega, O. Kupiainen-Määttä, A.P. Praplan, A. Adamov, A. Amorim, F. Bianchi, M. Breitenlechner, A. David, J. Dommen, N.M. Donahue, A. Downard, E. Dunne, J. Duplissy, S. Ehrhart, R.C. Flagan, A. Franchin, R. Guida, J. Hakala, A. Hansel, M. Heinritzi, H. Henschel, T. Jokinen, H. Junninen, M. Kajos, J. Kangasluoma, H. Keskinen, A. Kupc, T. Kurten, A.N. Kvashin, A. Laaksonen, K. Lehtipalo, M. Leiminger, J. Leppä, V. Loukonen, V. Makhmutov, S. Mathot, M.J. McGrath, T. Nieminen, T. Olenius, A. Onnela, T. Petäjä, F. Riccobono, I. Riipinen, M. Rissanen, L. Rondo, T. Ruuskanen, F.D. Santos, N. Sarnela, S. Schallhart, R. Schnitzhofer, J.H. Seinfeld, M. Simon, M. Sipilä, Y. Stozhkov, F. Stratmann, A. Tomé, J. Tröstl, G. Tsagkogeorgas, P. Vaattovaara, Y. Viisanen, A. Virtanen, A. Vrtala, P.E. Wagner, E. Weingartner, H. Wex, C. Williamson, D. Wimmer, P. Ye, T. Yli-Juuti, K.S. Carslaw, M. Kulmala, J. Curtius, U. Baltensperger, D.R. Worsnop, H. Vehkamäki, J. Kirkby, Molecular understanding of sulphuric acid-amine particle nucleation in the atmosphere, Nature 502 (2013) 359–363.

[13] O. Kupiainen, I.K. Ortega, T. Kurtén, H. Vehkamäki, Amine substitution into sulfuric acid – ammonia clusters, Atmos. Chem. Phys. 12 (2012) 3591-3599.

[14] K.D. Froyd, E.R. Lovejoy, Bond energies and structures of ammonia-sulfuric acid positive cluster ions, J. Phys. Chem. A 116 (2012) 5886-5899.

---

## Author Comment (AC2)

**Responses to Referee #2's comments**

We are grateful to the reviewers for their valuable and helpful comments on our manuscript **"Molecular-level study on the role of methanesulfonic acid in iodine oxoacids nucleation"** (MS No.: egusphere-2023-2084). We have revised the manuscript carefully according to reviewers' comments. The point-to-point responses to the Referee #2's comments are summarized below:

**Referee comments:**

Jing Li et al. reports a theoretical study on the iodic acid ($HIO_3$) – iodous acid ($HIO_2$) based nucleating process enhanced by methanesulfonic acid (MSA) by quantum chemical calculation and cluster dynamic simulation. They found that the MSA can enhance the $HIO_3$-$HIO_2$-based nucleation, especially in polar oceanic regions. This manuscript has systematically studied the $HIO_3$-$HIO_2$-MSA ternary nucleation system, covering cluster stability, thermodynamic/kinetic analysis, and molecular-level mechanism. These interesting findings show the significance of sulfur and iodine synergistic nucleation, providing deeper insights into marine secondary particle formation, given chemical complexity of real atmosphere. This well-written manuscript has important atmospheric implications, such as in the studies of marine aerosol formation and the sulfur/iodine cycling. Hence, I recommend the publication of this study in *Atmospheric Chemistry and Physics* after considering my comments listed below.

**Response:** We appreciate the reviewer for dedicating time to assess our manuscript and providing valuable comments and positive feedback.
* * *
**General comments:**

**Comment 1:** Although the authors have provided sufficient computational details, it would be better to add the grid settings for DFT calculations and the optimization convergence in method section of the main text.

**Response:** Based on the reviewer's suggestion, we have added the adopted grid settings (FineGrid) of DFT calculations in Section 2.1 in the main text. Accordingly, the corresponding sentence in Lines 64-65 of page 3 has been restructured as follows: "All density functional

theory (DFT) calculations were carried out using the Gaussian 09 package (Frisch et al., 2009), where FineGrid and tight convergence were employed."
* * *
**Comment 2:** In general, the nucleation process involves competition between cluster collision and its evaporation processes. In the ACDC simulations presented in this manuscript, could the authors specify which types of collision and evaporation processes considered? If possible, these details would be better to added in the ACDC methodology section.

**Response:** Thanks. The reviewer's professional comment is beneficial in enhancing the readers' understanding of more simulation details. Accordingly, the detailed settings about the collision and evaporation processes in ACDC simulations have been added in the revised manuscript (Lines 100-102, page 4) as follows: "In the performed ACDC simulations, all possible collision and evaporation processes, including monomer-monomer, monomer-cluster, cluster-cluster collisions, as well as the decomposition of parent clusters into monomers and clusters, or into two smaller clusters, were taken into account."
* * *
**Comment 3:** In this work, the authors systematically investigated the $HIO_3$-$HIO_2$-MSA ternary nucleation process, where $HIO_2$ appears to play a crucial role in all clustering pathways. How should this be interpreted?

**Response:** Thanks. It is indeed an important query. As expertly suggested by the reviewer, $HIO_2$ appears to play a crucial role in all clustering pathways of $HIO_3$-$HIO_2$-MSA nucleation. This is mainly because $HIO_2$, when interacting with acidic $HIO_3$ or MSA, behaves like base molecules[1-3]. Specifically, it can serve as a proton acceptor, being protonated by $HIO_3$ or MSA, leading to the formation of stable acid-base ion pairs. Once $HIO_2$ is missing, the acid-base reaction between $HIO_3$ and MSA cannot occur. Thus, $HIO_2$-induced acid-base reactions with MSA and $HIO_3$ yield ion pairs whose electrostatic interactions enhance the stability of the formed cluster.
* * *
**Comment 4:** For the formed $HIO_3$-$HIO_2$-MSA clusters, especially large-sized clusters, are there any unoccupied binding sites that can enable further molecular binding and cluster growth?

The authors would better provide theoretical evidence by quantum chemical calculations, such as using wave function analysis, or others. If done, this will provide the readers with an intuitive understanding of the growth potential of the cluster.

**Response:** Following the professional advice of the reviewer, we have performed the electrostatic potential (ESP) analysis for the vacant sites of the stable large-size structure cluster in the revised manuscript. The results of ESP analysis are presented in Fig. S2 below. The red localizations with maximum ESP at the end of the iodine and hydrogen atoms within $HIO_3$ and $HIO_2$ along the O–I and O–H direction, which can act as XB or HB donor sites. While the blue regions with minimum ESP of the terminal oxygen atoms have strong nucleophilicity as the HB or XB acceptors.

Herein, we have added the ESP results and analysis to the revised manuscript, and for the convenience of the review, we have copied Figure S2 and the corresponding analysis (Lines 129-132, page 5) as following: "Additionally, taking the $(HIO_3)_1(HIO_2)_3(MSA)_1$ cluster for example, there are still some potential remaining unoccupied binding sites as shown in Fig. S2. It suggests that the studied large-size clusters still have unoccupied HB and XB sites that can potentially facilitate the condensation of precursors in the atmosphere, enhancing further growth of marine aerosols."

[Figure]

unoccupied HB (or XB) acceptor site

unoccupied HB (or XB) donor site

$(HIO_3)_1(HIO_2)_3(MSA)_1$

HB: Hydrogen bond
XB: Halogen bond

**Figure S2.** The ESP-mapped molecular vdW surfaces of the $(HIO_3)_1(HIO_2)_3(MSA)_1$ cluster. The red region is the electron-deficient region, and the blue region is the electron-rich region.
* * *
**Specific comments:**

**Comment 5.**

**Page 3-4, Line 76 and 91:** In the equation (2) and (3), the operator $\sum_{i=1}^{n} N_i$, $\sum_{j<i} \beta$ should be change to:

$$\sum_{i=1}^{n} N_i \quad , \qquad \sum_{j<i} \beta$$

**Response:** The operators in the equation (2) and (3) have been corrected accordingly as suggested by the reviewer.
* * *
**Comment 6.**

**Page 4, Line 96:** Please provide more details of the calculations of the volume of cluster $i$ ($V_i$).

**Response:** According to the reviewer's suggestion, the details of the calculations of the volume of cluster $i$ ($V_i$) have been added in Lines 95-96 (page 4) of the revised manuscript as follows: "And $V_i = 3/4 \times \pi \times (d_i/2)^3$, where the diameter $d_i$ of cluster $i$ is derived from the cluster volume $V_i$ calculated by Multiwfn 3.7."
* * *
**Comment 7.**

**Page 5, Figure 1:** The authors seem to have forgotten to plot the hydrogen and halogen bonds with dotted lines in Fig. 1(d), please correct it.

**Response:** Thanks for the reviewer's carefulness. We have added the dotted lines indicating hydrogen and halogen bonds in Fig. 1(d) in the revised manuscript.
* * *
**Comment 8.**

**Page 6, Line 141:** In the sentence, 'Here, the condensation sink (CS) coefficient is set to be 2.0 $\times 10^{-3}$', there is a missing unit after the CS value.

**Response:** Thanks for the reviewer's careful reading. We have added the missing unit 's$^{-1}$' after the CS value in the revised manuscript (Line 152, page 6).
* * *
**Comment 9.**

**Page 7, Figure 3:** To enhance data clarity for readers, consider enlarging the font size in Fig. 3, which appears relatively small.

**Response:** Based on the reviewer's suggestion, we have adjusted the small font in Fig. 3 in the revised manuscript to enhance content clarity.
* * *
**Comment 10.**

**Page 12, Line 252:** The word 'play' should be 'plays'.

**Response:** Accordingly, the word 'play' has been changed to 'plays' in Line 280 of the revised manuscript.

**Reference:**

[1] L. Liu, S. Li, H. Zu, X. Zhang, Unexpectedly significant stabilizing mechanism of iodous acid on iodic acid nucleation under different atmospheric conditions, Sci. Total Environ. 859 (2023) 159832.
[2] S. Zhang, S. Li, A. Ning, L. Liu, X. Zhang, Iodous acid - a more efficient nucleation precursor than iodic acid, Phys. Chem. Chem. Phys. 24 (2022) 13651–13660.
[3] R. Zhang, H.B. Xie, F. Ma, J. Chen, S. Iyer, M. Simon, M. Heinritzi, J. Shen, Y.J. Tham, T. Kurtén, D.R. Worsnop, J. Kirkby, J. Curtius, M. Sipilä, M. Kulmala, X.C. He, Critical Role of Iodous Acid in Neutral Iodine Oxoacid Nucleation, Environ. Sci. Technol. 56 (2022) 14166–14177.

---

## Author Comment (AC3)

**Responses to Referee #1's comments**

We are grateful to the reviewers for their valuable and helpful comments on our manuscript "**Molecular-level study on the role of methanesulfonic acid in iodine oxoacids nucleation**" (MS No.: egusphere-2023-2084). We have revised the manuscript carefully according to reviewers' comments. The point-to-point responses to the Referee #1's comments are summarized below:

**Referee comments:**

Li et al. explored the role of methanesulfonic acid (MSA) in iodine oxoacids nucleation. Detailed molecular-level mechanisms of cluster formation were studied using quantum chemical methods and cluster dynamics, providing theoretical evidence for the contribution of MSA to the formation of marine iodine clusters. After carefully reading the manuscript, I find that the main argument that MSA may enhance the nucleation rate of iodine oxoacids is convincing. The contribution of MSA to the formation of marine iodine particles remains an open question because other acids and bases such as sulfuric acid and amines may also affect the $HIO_3$-$HIO_2$ nucleation process in the real atmosphere - as the authors have addressed at the end of this manuscript - while this study provides an important theoretical basis for this question. This manuscript is well written. I recommend it be accepted by *Atmospheric Chemistry and Physics*. A few minor comments on the interpretation of the theoretical results are given below.

**Response:** Thanks sincerely for the reviewer's professional and positive comments. We have revised the manuscript accordingly. The detailed point-to-point responses are listed as follows.
* * *
**General comments:**

**Comment 1:** Figures 1-3 seem to suggest that the significant enhancement of MSA on $HIO_3$-$HIO_2$ nucleation is robust against the uncertainties of cluster stability. However, would it be possible to have a supplementary or appendix figure for the general audience, showing the uncertainty range of the enhancement or relative contribution of MSA to the cluster formation rate?

**Response:** Thanks for the reviewer's professional and helpful comments. The uncertainties of enhancement of MSA on cluster formation may stem from ACDC simulations and quantum

chemical (QC) calculations, thereby we examined how variable ACDC settings, such as condensation sink coefficient (CS), sticking factor (SF, corresponding to a sticking probability for cluster/monomer collision) and the change of calculated Gibbs free energy of cluster formation ($\Delta G$, from quantum chemical calculations) impact the enhancement of MSA ($R_{MSA}$) on cluster formation rate ($J$). Here, the CS values ranged from $1.0 \times 10^{-4}$ $s^{-1}$ to $1.0 \times 10^{-2}$ $s^{-1}$, covering possible CS in relatively clean and polluted regions[1, 2]. The range of SF was set from 0.1 to 1.0 since sticking probabilities for neutral-neutral collisions between 0.1 and 1.0[3].

As shown in the Figs. S15 (a) and (b), although both CS and SF affect $R_{MSA}$ to some extent, the uncertainty range are relatively limited (CS < 32.5% and SF < 17.1%) and the results does not affect the trend and main conclusions.

[Figure]

**Figure S15.** Variation of enhancement strength $R$ of MSA with **(a)** condensation sink coefficient (CS) and **(b)** sticking factor (SF) for $HIO_3$-$HIO_2$-MSA system at $T = 278$ K, [$HIO_3$] $= 1.0 \times 10^7$, [$HIO_2$] $= 2.0 \times 10^5$, and [MSA] $= 1.0 \times 10^7$ molec. $cm^{-3}$.

In addition, the potential uncertainty of quantum chemical calculations is ultimately manifested in the calculated $\Delta G$ values. As reported by Kupiainen[4] et al. (2012), the differences between the computational (DFT//RI-CC2 method) and experimental $\Delta G$ values are about 1 kcal $mol^{-1}$ or less[5]. Accordingly, Almedia[3] et al. (2013) calculated the uncertainty range of ACDC simulated cluster formation resulting from QC calculations by adjusting the binding energy ($\pm 1$ kcal $mol^{-1}$). Further given the consistency of our research framework (DFT//RI-CC2 + ACDC) with Almedia et al. (2013), herein we have performed the uncertainty analysis of $R_{MSA}$ caused by QC calculations through adding or subtracting 1 kcal $mol^{-1}$ from the $\Delta G$ (using $\Delta G_{278K}$ as a reference). The figure below presents the uncertainty analysis results of

$J$ and $R_{MSA}$ at $T$ = 278 K, CS = $2.0 \times 10^{-3}$ s$^{-1}$, [HIO$_3$] = $10^7$, [HIO$_2$] = $2.0 \times 10^5$, [MSA] = $10^6$ –

$10^8$ molec. cm$^{-3}$.

[Figure]

**Figure S16.** Cluster formation rate $J$ **(a)** and enhancement strength $R$ of MSA **(b)** as a function

of [MSA] = $10^6$ – $10^8$ molec. cm$^{-3}$, with different energy of $\Delta G_{278K}$ (black line), $\Delta G_{278K}$ + 1

(blue line), $\Delta G_{278K}$ – 1 (red line), at $T$ = 278 K, CS = $2.0 \times 10^{-3}$ s$^{-1}$, [HIO$_3$] = $10^7$, [HIO$_2$] = 2.0

$\times 10^5$ molec. cm$^{-3}$.

Here, we have added the results of $R_{MSA}$ under different CS, SF and $\Delta G$ to the revised

supporting file, and for the convenience of the review, we have copied Figures S15-S16 and

the corresponding analysis as following: "Here, the potential uncertainties may stem from

ACDC simulations and quantum chemical (QC) calculations, we examined the effect of

condensation sink coefficient (CS), sticking factor (SF) and calculated $\Delta G$ of clusters on

enhancement of MSA to the cluster formation rate. The CS values ranged from $1.0 \times 10^{-4}$ s$^{-1}$ to

$1.0 \times 10^{-2}$ s$^{-1}$, covering possible CS in relatively clean and polluted regions[1, 2]. The range of

SF was set from 0.1 to 1.0 since sticking probabilities for neutral-neutral collisions between 0.1

and 1.0[3]. Both the CS and SF slightly affect the enhancement of MSA, with limited uncertainty

range of CS < 32.5% and SF < 17.1% (Fig. S15). As reported by Kupiainen[4] et al. (2012), the

differences between the computational (DFT//RI-CC2 method) and experimental $\Delta G$ values are

about 1 kcal mol$^{-1}$ or less[5]. Accordingly, Almedia[3] et al. (2013) calculated the uncertainty

range of ACDC simulated cluster formation resulting from QC calculations by adjusting the

binding energy ($\pm$1 kcal mol$^{-1}$). Further given the consistency of our research framework

(DFT//RI-CC2 + ACDC) with Almedia et al. (2013), herein we have performed the uncertainty

analysis of $R_{MSA}$ caused by QC calculations through adding or subtracting 1 kcal mol$^{-1}$ from the $\Delta G$ (using $\Delta G_{278K}$ as a reference). As shown in Table S8 and Fig. S16, adjusting the $\Delta G_{278K}$ of clusters by ±1 kcal mol$^{-1}$ resulted in a minor variation in $J$ and $R$ of MSA, with the overall trend remaining consistent."
* * *
**Comment 2:** I found it challenging to interpret the relative importance of the MSA-involved path in HIO$_3$-HIO$_2$-MSA nucleation. Figure 3a shows that the MSA-involved path is a major path (74 %), yet this was simulated with a [MSA] 5 times of [HIO$_3$]. With the same [MSA] and [HIO$_3$], the MSA-involved path was expected to contribute ~20 %, showing that MSA was a bit less efficient than HIO$_3$ in clustering with HIO$_2$. This comparatively lower efficiency does not affect the main conclusion as the [MSA] may exceed [HIO$_3$] in atmospheric environments. However, Figures 4 and 5 show a high enhancement factor (> 2) with the same [MSA] and [HIO$_3$]. This high enhancement factor indicates that MSA is more efficient than what I interpreted above. I hope this can be clarified in the revised manuscript.

**Response:** This is a very insightful point – thanks for bringing it up. Indeed, as expertly assessed by the reviewer, HIO$_3$ undergo clustering with HIO$_2$ more efficiently than MSA due to the lower contribution of MSA-involved pathway (~20%) at same concentrations of HIO$_3$ and MSA ($10^7$ molec. cm$^{-3}$). However, in this case, the involvement of MSA in nucleation shows a high enhancement factor (> 2) for rate $J$, which is indeed the point that may confuse the reader. Accordingly, we explored the underlying nucleation mechanism under the focused condition: $T = 278K$, CS $= 2.0 \times 10^{-3}$ s$^{-1}$, (a) [HIO$_3$] = [MSA] = $1.0 \times 10^7$, and [HIO$_2$] = $2.0 \times 10^5$ molec. cm$^{-3}$. (b) [HIO$_3$] = [MSA] = $1.0 \times 10^6$, and [HIO$_2$] = $2.0 \times 10^4$ molec. cm$^{-3}$.

As shown in Fig. S7, the contribution of MSA to clustering consists not only of directly forming HIO$_3$-HIO$_2$-MSA clusters (~20%), but also its 'catalysis' role in facilitating formation of initial HIO$_3$-HIO$_2$ clusters, e.g., (HIO$_3$)$_1$(HIO$_2$)$_{1-2}$, through a process of first participation in forming the (HIO$_3$)$_1$(HIO$_2$)$_{1-2}$(MSA)$_1$ clusters, and then evaporation out. Taken together, MSA promotes both HIO$_3$-HIO$_3$-MSA and HIO$_3$-HIO$_2$ clustering pathways, and its dual contribution results in a high enhancement factor (> 2).

Furthermore, to make the readers clear, we accordingly provide an explanatory account of this phenomenon as follows (Lines 227-231 in the revised manuscript): "However, the

atmospheric [HIO$_3$] ranges widely from $10^6$ to $10^8$ molec. cm$^{-3}$. When [HIO$_3$] is comparable or higher than [MSA], the HIO$_3$-HIO$_2$ pathway contributes more, and the $R$ of MSA decreases with the rising [HIO$_3$]. It is worth noting that when [HIO$_3$] is comparable to [MSA], the $R$ of MSA is greater than 2, as the contribution of MSA to clustering includes not only the direct formation of HIO$_3$-HIO$_2$-MSA clusters (~20%), but also its 'catalysis' role in facilitating formation of initial HIO$_3$-HIO$_2$ clusters (Fig. S7)."

[Figure]

**Figure S7.** Main cluster growth pathway of the HIO$_3$-HIO$_2$-MSA nucleating system at $T$ = 278K, CS = $2.0 \times 10^{-3}$ s$^{-1}$, (a) [HIO$_3$] = $1.0 \times 10^7$, [HIO$_2$] = $2.0 \times 10^5$, and [MSA] = $1.0 \times 10^7$ molec. cm$^{-3}$, (b) [HIO$_3$] = $1.0 \times 10^6$, [HIO$_2$] = $2.0 \times 10^4$, and [MSA] = $1.0 \times 10^6$ molec. cm$^{-3}$.
* * *
**Specific comments:**

**Comment 3:** Line 180, "Overall, the results suggest that MSA's contribution to cluster formation is positively related to [MSA] but negatively linked to [$HIO_3$]." This sentence is correct in terms of the relative contribution but awkward. How about removing this sentence and adding discussions on the relative importance of the MSA path (see comment 2)?

**Response:** According to the reviewer's helpful suggestion, we have removed the mentioned sentence "Overall, the results suggest that MSA's contribution to cluster formation is positively related to [MSA] but negatively linked to [$HIO_3$]" in Line 180, and added the discussion of comment 2 in the revised manuscript (Lines 227-231, page 10).
* * *
**Comment 4:** Line 207, "To sum up, MSA can promote nucleation, particularly in marine regions characterized by lower T, lower [$HIO_3$] and [$HIO_2$]." I was confused that MSA can promote nucleation at low [$HIO_2$] within the context of this manuscript, as [$HIO_2$] is the starting point of cluster formation. This might be caused by overemphasizing the relative contribution. Also, [$HIO_2$] is usually associated with [$HIO_3$]. Replacing [$HIO_3$] (implicitly indicated to be independent of [$HIO_2$]) with [$HIO_3$]/[$HIO_2$] in some discussions may help with understanding.

**Response:** Certainly, as predicted by the reviewer, [$HIO_2$] is the starting point of cluster formation. When [$HIO_2$] declines, both MSA-involved and non-MSA pathway proportions decrease (see Fig. (3) in the main text). However, in this study, we draw the following conclusions based on the calculated enhancement strength ($R$) of MSA on nucleation, i.e., $R = J(HIO_3\text{-}HIO_2\text{-}MSA) /J(HIO_3\text{-}HIO_2)$. As [$HIO_2$] declines, both the numerator and denominator of $R$ decrease. However, the numerator diminishes relatively slowly due to the introduction of MSA, enhancing the rate and retarding its decay, which results in an increased $R$ value. Yet, in environments with higher [$HIO_2$], the enhancing effect of MSA will be weaker because the more efficient $HIO_3$ will fully combine with $HIO_2$, thereby resulting in a lower $R$.

  Accordingly, following the professional advice of the reviewer, to make it clear to the readers, we add the explanation prior to the introduction of the ACDC simulations as follows: "…where [$HIO_3$]/[$HIO_2$] is a constant" in Line 150 of page 6.
* * *
**Comment 5:** Line 222, "observed $J$ of $2.1 \times 10^{-4}$ cm$^{-3}$ s$^{-1}$". This $J$ value is too low from a

measurement point of view. In a scatter plot showing the correlation between $J$ and precursor concentrations, some small values of $J$ are often given, though they might be obtained during weak or non-NPF periods. I checked the SI of Beck et al. and found that they have clarified that "Data with $J$-values < a few $10^{-3}$ $cm^{-3}$ $s^{-1}$ are highly unreliable and reflect mainly the noise levels...". Figure 2a in their main manuscript shows that the $J$ value is ~0.1 $cm^{-3}$ $s^{-1}$ during typical NPF events.

**Response:** Thanks for these professional and rigorous suggestions. Accordingly, given the unreliability of $J$-values (< a few $10^{-3}$ $cm^{-3}$ $s^{-1}$), we have adjusted the range of the observed cluster formation rate $J$ of Ny-Ålesund to $10^{-3}$ – $10^{-1}$ $cm^{-3}$ $s^{-1}$ in Fig. 6(a). And the sentence "…the $J(HIO_3-HIO_2-MSA)$ can be two orders of magnitude higher than the observed $J$ of $2.1 \times 10^{-4}$ $cm^{-3}$ $s^{-1}$" has been changed to "…the $J(HIO_3-HIO_2-MSA)$ can be one order of magnitude higher than the observed $J$ of $10^{-3}$ $cm^{-3}$ $s^{-1}$" in Lines 248-249 (page 11) of the revised manuscript.
* * *
**Comment 6:** Figure 6. It is recommended to adjust the shaded area of field observation. Now the measured $J$ shares a similar style with the simulated $J$. Some readers may wonder why there is no correlation between the measured $J$ and $[HIO_3]$.

**Response:** Thanks. The reviewers' suggestion is helpful in improving the clarity of the data. Accordingly, we have changed the measured $J$ with a different style in Fig. 6 from shaded areas to dashed lines.
* * *
**Comment 7:** Figure 6 caption. Please explain why there is a single line for $HIO_3-HIO_2$ while $[HIO_2]$ ranges from 2e3 to 2e4. Is it because of a constant $[HIO_2]/[HIO_3]$ in the simulation?

**Response:** As the reviewer's expert insight suggests, the reason that the rate of $HIO_3-HIO_2$ cluster formation presents as a line that increases with $[HIO_3]$ is that $[HIO_3]/[HIO_2]$ is fixed to a constant value (50, according to the field measured ratio from Sipilä et al. 2016[6]) in the simulation. To make it clearer for the readers, we have added a description of the relationship between $[HIO_3]$ and $[HIO_2]$ to the caption of Fig.6 as: "$[HIO_3]/[HIO_2]$ is a constant".
* * *
**Comment 8:** Figure S5. Please give the reference for field observation data herein. It is surprising to see a high formation rate of 1e4-1e6 $cm^{-3}$ $s^{-1}$.

**Response:** According to the reviewer's suggestion, the corresponding reference (O'Dowd, et al., *J. Geophys. Res.: Atmos.,* 2002)[7] has been cited in the caption of Figure S14 (previous Figure S5) in the revised supporting information.
* * *
**Comment 9:** Line 248, "...thermodynamic analyses suggest that MSA-involved clustering is nearly barrierless". I do not disagree with this statement, yet it may confuse some readers, especially considering that the horizontal axes in Figs. 5-6 are [$HIO_3$] rather than [$HIO_2$]. How about emphasizing that the $HIO_2$ addition, as the rate-limiting step for cluster formation and growth, is nearly barrierless?

**Response:** Thanks for the reviewer's professional comments. The preference for [$HIO_3$] as the horizontal axes in Figs. 5-6 is due to the strong correlation between NPF occurrence and observations of iodic acid[6]. Also, in CLOUD experiments, the nucleation rates show a strong dependency on $HIO_3$ concentration[8]. Conversely, there are limited available field observations of $HIO_2$, characterized by lower reported concentrations, despite its pivotal role in stabilizing $HIO_3$ cluster. Therefore, the concentration of $HIO_3$ is employed here as horizontal axes to present the results of cluster formation rate or enhancement strength.

Furthermore, as mentioned by the reviewer, the $HIO_2$ addition is the rate-limiting step for cluster formation, which leads to the significant increasement of the $J$($HIO_3$-$HIO_2$-MSA) compared to $J$($HIO_3$-MSA) (Figure S12). As shown in Figure S13, thermodynamic analysis suggest that compared with $HIO_3$-MSA pathway, $HIO_3$-$HIO_2$-MSA path is almost barrierless (1.24 kcal mol$^{-1}$) at $T = 278$ K, [$HIO_3$] $= 1.0 \times 10^6$, [$HIO_2$] $= 2.0 \times 10^4$, and [MSA] $= 5.0 \times 10^6$ molec. cm$^{-3}$, indicating that the $HIO_2$ addition is favorable. To make the readers clear, we accordingly copy the explanation as follows (Lines 235-238 in the revised manuscript): "Furthermore, the effect of $HIO_2$ addition on the whole nucleation system was considered, as it is not only the rate-limiting step for cluster formation, leading to the significant increasement of the $J$($HIO_3$-$HIO_2$-MSA) compared to $J$($HIO_3$-MSA) (Figure S12), but also thermodynamically favorable due to $HIO_3$-$HIO_2$-MSA path is almost barrierless (1.24 kcal mol$^{-1}$) compared to $HIO_3$-MSA pathway (Figure S13)."

[Figure]

**Figure S12.** Simulated cluster formation rates $J$ (cm$^{-3}$ s$^{-1}$) against varying [MSA] = $10^6$ – $10^7$ molec. cm$^{-3}$, at $T$ = 278 K, CS = 2.0 × 10$^{-3}$ s$^{-1}$, [HIO$_3$] = 1.0 × 10$^6$, [HIO$_2$] = 2.0 × 10$^4$ molec. cm$^{-3}$.

[Figure]

**Figure S13.** The Gibbs free energies of cluster formation ($\Delta G$, kcal mol$^{-1}$) based on the main clustering pathway in HIO$_3$-MSA and HIO$_3$-HIO$_2$-MSA nucleation system at $T$ = 278 K, CS = 2.0 × 10$^{-3}$ s$^{-1}$, [HIO$_3$] = 1.0 × 10$^6$, [HIO$_2$] = 2.0 × 10$^4$ molec. cm$^{-3}$.

**Reference:**

[1] X.-C. He, Y.J. Tham, L. Dada, M. Wang, H. Finkenzeller, D. Stolzenburg, S. Iyer, M. Simon, A.K. ˙´rten, J. Shen, B. Roerup, M. Rissanen, S. Schobesberger, R. Baalbaki, D.S. Wang, T.K. Koenig, T. Jokinen, N. Sarnela, L.J. Beck, J. Almeida, S. Amanatidis, A. Amorim, F. Ataei, A. Baccarini, B. Bertozzi, F. Bianchi, S. Brilke, L. Caudillo, D. Chen, R. Chiu, B. Chu, A. Dias, A. Ding, J. Dommen, J. Duplissy, I.E. Haddad, L.G. Carracedo, M. Granzin, A. Hansel, M. Heinritzi, V. Hofbauer, H. Junninen, J. Kangasluoma, D. Kemppainen, C. Kim, W. Kong, J.E. Krechmer, A. Kvashin, T. Laitinen, H. Lamkaddam, C.P. Lee, K. Lehtipalo, M. Leiminger, Z. Li, V. Makhmutov, H.E. Manninen, G. Marie, R. Marten, S. Mathot, R.L. Mauldin, B. Mentler, O. Moehler, T. Mueller, W. Nie, A. Onnela, T. Petaja, J. Pfeifer, M. Philippov, A. Ranjithkumar, A. Saiz-Lopez, I. Salma, W. Scholz, S. Schuchmann, B. Schulze, G. Steiner, Y. Stozhkov, C. Tauber, A. Tome, R.C. Thakur, O. Vaisanen, M. Vazquez-Pufleau, A.C. Wagner, Y. Wang, S.K. Weber, P.M. Winkler, Y. Wu, M. Xiao, C. Yan, Q. Ye, A. Ylisirnio, M. Zauner-Wieczorek, Q. Zha, P. Zhou, R.C. Flagan, J. Curtius, U. Baltensperger, M. Kulmala, V.-M. Kerminen, T. Kurten, N.M. Donahue, R. Volkamer, J. Kirkby, D.R. Worsnop, M. Sipila, Role of iodine oxoacids in atmospheric aerosol nucleation, Science 371 (2021) 589–595.

[2] H. Yu, L. Ren, X. Huang, M. Xie, J. He, H. Xiao, Iodine speciation and size distribution in ambient aerosols at a coastal new particle formation hotspot in China, Atmos. Chem. Phys. 19 (2019) 4025–4039.

[3] J. Almeida, S. Schobesberger, A. Kürten, I.K. Ortega, O. Kupiainen-Määttä, A.P. Praplan, A. Adamov, A. Amorim, F. Bianchi, M. Breitenlechner, A. David, J. Dommen, N.M. Donahue, A. Downard, E. Dunne, J. Duplissy, S. Ehrhart, R.C. Flagan, A. Franchin, R. Guida, J. Hakala, A. Hansel, M. Heinritzi, H. Henschel, T. Jokinen, H. Junninen, M. Kajos, J. Kangasluoma, H. Keskinen, A. Kupc, T. Kurten, A.N. Kvashin, A. Laaksonen, K. Lehtipalo, M. Leiminger, J. Leppä, V. Loukonen, V. Makhmutov, S. Mathot, M.J. McGrath, T. Nieminen, T. Olenius, A. Onnela, T. Petäjä, F. Riccobono, I. Riipinen, M. Rissanen, L. Rondo, T. Ruuskanen, F.D. Santos, N. Sarnela, S. Schallhart, R. Schnitzhofer, J.H. Seinfeld, M. Simon, M. Sipilä, Y. Stozhkov, F. Stratmann, A. Tomé, J. Tröstl, G. Tsagkogeorgas, P. Vaattovaara, Y. Viisanen, A. Virtanen, A. Vrtala, P.E. Wagner, E. Weingartner, H. Wex, C. Williamson, D. Wimmer, P. Ye, T. Yli-Juuti, K.S. Carslaw, M. Kulmala, J. Curtius, U. Baltensperger, D.R. Worsnop, H. Vehkamäki, J. Kirkby, Molecular understanding of sulphuric acid-amine particle nucleation in the atmosphere, Nature 502 (2013) 359–363.

[4] O. Kupiainen, I.K. Ortega, T. Kurtén, H. Vehkamäki, Amine substitution into sulfuric acid – ammonia clusters, Atmos. Chem. Phys. 12 (2012) 3591-3599.

[5] K.D. Froyd, E.R. Lovejoy, Bond energies and structures of ammonia-sulfuric acid positive cluster ions, J. Phys. Chem. A 116 (2012) 5886-5899.

[6] M. Sipilä, N. Sarnela, T. Jokinen, H. Henschel, H. Junninen, J. Kontkanen, S. Richters, J. Kangasluoma, A. Franchin, O. peräkylä, M.P. Rissanen, M. Ehn, H. Vehkamäki, T. Kurten, T. Berndt, T. Petäjä, D. Worsnop, D. Ceburnis, V.M. Kerminen, M. Kulmala, C. O'Dowd, Molecular-scale evidence of aerosol particle formation via sequential addition of $HIO_3$, Nature 537 (2016) 532–534.

[7] C.D. O'Dowd, K. Hämeri, J. Mäkelä, M. Väkeva, P. Aalto, G. de Leeuw, G.J. Kunz, E. Becker, H.C. Hansson, A.G. Allen, R.M. Harrison, H. Berresheim, C. Kleefeld, M. Geever, S.G. Jennings, M. Kulmala, Coastal new particle formation: Environmental conditions and aerosol physicochemical characteristics during nucleation bursts, J. Geophys. Res.-Atmos. 107 (2002).

[8] X.-C. He, S. Iyer, M. Sipilä, A. Ylisirniö, M. Peltola, J. Kontkanen, R. Baalbaki, M. Simon, A. Kürten, Y.J. Tham, J. Pesonen, L.R. Ahonen, S. Amanatidis, A. Amorim, A. Baccarini, L. Beck, F. Bianchi, S. Brilke, D. Chen, R. Chiu, J. Curtius, L. Dada, A. Dias, J. Dommen, N.M. Donahue, J. Duplissy, I. El

Haddad, H. Finkenzeller, L. Fischer, M. Heinritzi, V. Hofbauer, J. Kangasluoma, C. Kim, T.K. Koenig, J. Kubečka, A. Kvashnin, H. Lamkaddam, C.P. Lee, M. Leiminger, Z. Li, V. Makhmutov, M. Xiao, R. Marten, W. Nie, A. Onnela, E. Partoll, T. Petäjä, V.-T. Salo, S. Schuchmann, G. Steiner, D. Stolzenburg, Y. Stozhkov, C. Tauber, A. Tomé, O. Väisänen, M. Vazquez-Pufleau, R. Volkamer, A.C. Wagner, M. Wang, Y. Wang, D. Wimmer, P.M. Winkler, D.R. Worsnop, Y. Wu, C. Yan, Q. Ye, K. Lehtinen, T. Nieminen, H.E. Manninen, M. Rissanen, S. Schobesberger, K. Lehtipalo, U. Baltensperger, A. Hansel, V.-M. Kerminen, R.C. Flagan, J. Kirkby, T. Kurtén, M. Kulmala, Determination of the collision rate coefficient between charged iodic acid clusters and iodic acid using the appearance time method, Aerosol Sci. Tech. 55 (2021) 231-242.

---

## Author Comment (AC4)

**Responses to Referee #4's comments**

We sincerely appreciate the reviewer's valuable and helpful comments on our manuscript **"Molecular-level study on the role of methanesulfonic acid in iodine oxoacids nucleation"** (MS No.: egusphere-2023-2084). We have revised the manuscript carefully according to reviewer's comments. The point-to-point responses to the Referee #4's comments are summarized below:

**Referee comments:**

This manuscript explores the enhancement effects of MSA on the $HIO_3$-$HIO_2$ system through DFT calculations and kinetic analysis. It is found that adding MSA significantly enhances the nucleation rate, especially in colder regions. The calculations are also compared with observations and in general the MSA-$HIO_3$-$HIO_2$ nucleation better explains the results than the binary $HIO_3$-$HIO_2$ nucleation. Overall, I find this manuscript clearly written and it is a nice contribution to the literature. The manuscript can be published after the following comments are addressed.

**Response:** We would like to thank the reviewer for taking the time to review our manuscript and for providing the professional comments and positive feedback.
* * *
**Major comments:**

**Comment 1:** What is the criterion for stable clusters (based on which nucleation rates can be defined)? Are they determined based on the growth rate/dissociation rate ratio? A clear definition should be given in the main text.

**Response:** Indeed, as expertly suggested by the reviewer, whether a cluster is stable or not is determined based on the cluster growth rate/dissociation rate ratio. In ACDC simulations[1], stable clusters are those in which collisions with molecules can be assumed to dominate over cluster evaporation.

According to the reviewer's helpful suggestions, the corresponding definition has been added in the main text of the revised manuscript (Lines 102-103, Page 4) as follows: "Additionally, whether the clusters in the simulated system are stable depends on whether the rate of collision frequencies exceeds the total evaporation rate coefficients ($\beta C/\Sigma\gamma > 1$) (Table

S4)."
* * *
**Comment 2:** Section 3.4: how *J* is defined/calculated in the measurements should be discussed here, since I assume it is different from the definition used in the simulation. In other words, more justification of why the simulated rates and observed rates are directly comparable should be provided.

**Response:** This is a very helpful point – thanks for bringing it up. In the ACDC simulation, nucleation generally refers to the formation of relatively stable clusters for which collisions with molecules can be assumed to dominate over cluster evaporation. Accordingly, the cluster formation rate ($J$) indicates the particle flux out of the studied system. In this case, it is the rate of clusters forming at some specific size (*i.e.* the net flux into the size from all other sizes)[2]. In field observation, the formation rates ($J_{1.5}$) were measured by instruments, such as nitrate chemical ionization atmospheric pressure interface Time-Of-Flight mass spectrometer (CI-APi-TOF)[3], differential mobility particle sizer (DMPS) and neutral cluster and air ion spectrometer (NAIS)[4].

According to the Kerminen-Kulmala equation [5], cluster formation rates for $d_2$ nm clusters ($J_{d_2}$) relate to those for $d_1$ nm clusters ($J_{d_1}$) by

$$J_{d_1} = J_{d_2} \exp\left\{\gamma \left(\frac{1}{d_1} - \frac{1}{d_2}\right) \frac{\mathrm{CS}}{\mathrm{GR}_{d_2 \text{-} d_1}}\right\},$$

where the $\mathrm{GR}_{d_2\text{-}d_1}$ is the initial cluster growth rate from $d_1$ to $d_2$ nm, and CS represents condensation sink of clusters by preexisting particles. The parameter $\gamma$ depends on many factors but can usually be approximated by assuming it to be equal to 0.23 $\mathrm{nm^2\ m^2\ h^{-1}}$.

In this study, the relationship between the formation rates of simulated clusters ($J_{1.2}$) and that of observed clusters ($J_{1.5}$) can be written as:

$$J_{1.2} = J_{1.5} \exp\left\{0.23 \times \left(\frac{1}{1.2} - \frac{1}{1.5}\right) \frac{\mathrm{CS}}{\mathrm{GR}}\right\},$$

where GR was measured to be 3.2 – 4.4 $\mathrm{nm \cdot h^{-1}}$ in the 1.1 – 2.0 nm size range during three observed events[6, 7], and CS was 0.002 $\mathrm{s^{-1}}$. $J_{1.2}$ was then calculated to be 1.00001 – 1.00002 times of $J_{1.5}$. Thus, the observed cluster formation rates for 1.5 nm clusters can be directly comparable with the simulated $J_{1.2}$.

We have included corresponding justification in Section 3.4 of the revised manuscript

(Lines 242-243, Page 11) and supporting file as follows: "Subsequently, we compared these simulation results with observed nucleation rates and the definition of cluster formation rate was detailed in Supporting Information (SI)."
* * *
**Comment 3:** I suggest the authors do two types of calculations corresponding to polluted (CS larger than 0.002/s) and relatively clean environments. This could benefit future research in more polluted coastal regions.

**Response:** Thanks, these suggestions from the reviewer are very important for improving the environmental impacts of the $HIO_3$-MSA-$HIO_2$ nucleation. Accordingly, we have performed additional ACDC simulations with CS values of $1.0 \times 10^{-2}$ $s^{-1}$ and $1.0 \times 10^{-4}$ $s^{-1}$ corresponding to polluted and relatively clean environments, respectively. The figures below present the results of the simulated cluster formation rates $J$ (Figures S8-S9) and enhancement strength $R$ of MSA (Figures S10-S11).

[Figure]

**Figure S8.** Simulated cluster formation rates $J$ ($cm^{-3}$ $s^{-1}$) against varying atmospheric temperatures ($T = 258 - 298$ K), CS = $1.0 \times 10^{-2}$ $s^{-1}$, [$HIO_3$] = $1.0 \times 10^7$, [$HIO_2$] = $2.0 \times 10^5$, and [MSA] = $1.0 \times 10^7$ molec. $cm^{-3}$.

[Figure]

**Figure S9.** Simulated cluster formation rates $J$ (cm$^{-3}$ s$^{-1}$) against varying atmospheric temperatures ($T$ = 258 – 298 K), CS = 1.0 × 10$^{-4}$ s$^{-1}$, [HIO$_3$] = 1.0 × 10$^7$, [HIO$_2$] = 2.0 × 10$^5$, and [MSA] = 1.0 × 10$^7$ molec. cm$^{-3}$.

[Figure]

**Figure S10.** Enhancement strength $R$ of MSA on cluster formation rates at varying precursor concentrations: [HIO$_3$] = 10$^6$ – 10$^8$, [HIO$_2$] = 2.0 × 10$^4$ – 2.0 × 10$^6$ molec. cm$^{-3}$, **(a)** [MSA] = 1.0 × 10$^6$ molec. cm$^{-3}$, **(b)** [MSA] = 1.0 × 10$^7$ molec. cm$^{-3}$, and **(c)** [MSA] = 1.0 × 10$^8$ molec. cm$^{-3}$, $T$ = 278 K, CS = 1.0 × 10$^{-2}$ s$^{-1}$.

[Figure]

**Figure S11.** Enhancement strength $R$ of MSA on cluster formation rates at varying precursor

concentrations: $[HIO_3] = 10^6 - 10^8$, $[HIO_2] = 2.0 \times 10^4 - 2.0 \times 10^6$ molec. cm$^{-3}$, **(a)** [MSA] = $1.0 \times 10^6$ molec. cm$^{-3}$, **(b)** [MSA] = $1.0 \times 10^7$ molec. cm$^{-3}$, and **(c)** [MSA] = $1.0 \times 10^8$ molec. cm$^{-3}$, $T$ = 278 K, CS = $1.0 \times 10^{-4}$ s$^{-1}$.

Herein, we have added the simulated $J$ and $R$ results, along with their analysis, in the revised supporting file (Figures S8 – S11). For the convenience of the review, we have copied the corresponding analysis (Lines 233-235, Page 10) as following: "In addition, we also examined the conditions in relatively polluted (CS = $1.0 \times 10^{-2}$ s$^{-1}$) and clean environments (CS = $1.0 \times 10^{-4}$ s$^{-1}$) and found that, similar to the environment with CS value of $2.0 \times 10^{-3}$ s$^{-1}$, MSA exhibits significant promoting effects on iodine particle formation (Figs. S8-S11)."
* * *
**Comment 4:** Figure 6. I believe the blue line should be an area as the other two. Also, how does the rates differ if the uncertainties of the DFT calculations for key clusters are considered? A table might be provided for this uncertainty analysis.

**Response:** We appreciate the insightful and rigorous comments from the reviewer. These suggestions can enhance the robustness of the simulation results. The blue line in Fig. 6 depicts $HIO_3$-$HIO_2$ nucleation rate, and since the ratio $[HIO_3]/[HIO_2]$ is held constant (50) according to the measured ratio $HIO_3/HIO_2$ from Sipilä et al. 2016[8], resulting in a line increasing with $HIO_3$ concentration. We have added description of the relationship between $[HIO_3]$ and $[HIO_2]$ to the caption of Figure 6 as: "$[HIO_3]/[HIO_2]$ is a constant". Therefore, as $[HIO_3]$ increases, the $J(HIO_3$-$HIO_2)$ does change as a line.

In addition, following the expert advice of the reviewer, we examined the effects of DFT computational uncertainty for key clusters on the rate as well as on the enhancement $R$ of MSA. The uncertainties of the DFT calculations ultimately manifested in the calculated $\Delta G$ values. As reported by Kupiainen[9] et al. (2012), the differences between the computational (DFT//RI-CC2 method) and experimental $\Delta G$ values are about 1 kcal mol$^{-1}$ or less[10]. Accordingly, Almedia[11] et al. (2013) calculated the uncertainty range of ACDC simulated cluster formation resulting from QC calculations by adjusting the binding energy ($\pm$1 kcal mol$^{-1}$). Further given the consistency of our research framework (DFT//RI-CC2 + ACDC) with Almedia et al. (2013), herein we have performed the uncertainty analysis of $R_{MSA}$ caused by QC calculations through adding or subtracting 1 kcal mol$^{-1}$ from the $\Delta G$ (using $\Delta G_{278K}$ as a reference). The table and

figure below present the uncertainty analysis results of $J$ and $R_{MSA}$ at $T = 278$ K, CS $= 2.0 \times 10^{-3}$ s$^{-1}$, [HIO$_3$] $= 10^7$, [HIO$_2$] $= 2.0 \times 10^5$, [MSA] $= 10^6 - 10^8$ molec. cm$^{-3}$.

**Table S8.** Cluster formation rate $J$ of HIO$_3$-HIO$_2$-MSA system under different Gibbs free energy ($\Delta G_{278K}$, $\Delta G_{278K} + 1$, $\Delta G_{278K} - 1$) at $T = 278$ K, CS $= 2.0 \times 10^{-3}$ s$^{-1}$, [HIO$_3$] $= 10^7$, [HIO$_2$] $= 2.0 \times 10^5$, [MSA] $= 10^6 - 10^8$ molec. cm$^{-3}$.

| [MSA] | $J_{\Delta G}$ | $J_{\Delta G-1}$ | $J_{\Delta G+1}$ |
|---|---|---|---|
| $1.00 \times 10^6$ | $5.13 \times 10^0$ | $4.31 \times 10^0$ | $5.29 \times 10^0$ |
| $1.27 \times 10^6$ | $5.35 \times 10^0$ | $4.46 \times 10^0$ | $5.53 \times 10^0$ |
| $1.62 \times 10^6$ | $5.64 \times 10^0$ | $4.65 \times 10^0$ | $5.84 \times 10^0$ |
| $2.07 \times 10^6$ | $6.01 \times 10^0$ | $4.90 \times 10^0$ | $6.25 \times 10^0$ |
| $2.64 \times 10^6$ | $6.50 \times 10^0$ | $5.22 \times 10^0$ | $6.78 \times 10^0$ |
| $3.36 \times 10^6$ | $7.14 \times 10^0$ | $5.63 \times 10^0$ | $7.47 \times 10^0$ |
| $4.28 \times 10^6$ | $7.99 \times 10^0$ | $6.17 \times 10^0$ | $8.40 \times 10^0$ |
| $5.46 \times 10^6$ | $9.12 \times 10^0$ | $6.87 \times 10^0$ | $9.64 \times 10^0$ |
| $6.95 \times 10^6$ | $1.06 \times 10^1$ | $7.80 \times 10^0$ | $1.13 \times 10^1$ |
| $8.86 \times 10^6$ | $1.27 \times 10^1$ | $9.04 \times 10^0$ | $1.36 \times 10^1$ |
| $1.13 \times 10^7$ | $1.56 \times 10^1$ | $1.07 \times 10^1$ | $1.68 \times 10^1$ |
| $1.44 \times 10^7$ | $1.96 \times 10^1$ | $1.30 \times 10^1$ | $2.14 \times 10^1$ |
| $1.83 \times 10^7$ | $2.53 \times 10^1$ | $1.61 \times 10^1$ | $2.78 \times 10^1$ |
| $2.34 \times 10^7$ | $3.35 \times 10^1$ | $2.05 \times 10^1$ | $3.71 \times 10^1$ |
| $2.98 \times 10^7$ | $4.55 \times 10^1$ | $2.68 \times 10^1$ | $5.07 \times 10^1$ |
| $3.79 \times 10^7$ | $6.31 \times 10^1$ | $3.60 \times 10^1$ | $7.08 \times 10^1$ |
| $4.83 \times 10^7$ | $8.93 \times 10^1$ | $4.95 \times 10^1$ | $1.01 \times 10^2$ |
| $6.16 \times 10^7$ | $1.29 \times 10^2$ | $6.98 \times 10^1$ | $1.46 \times 10^2$ |
| $7.85 \times 10^7$ | $1.88 \times 10^2$ | $1.01 \times 10^2$ | $2.13 \times 10^2$ |
| $1.00 \times 10^8$ | $2.78 \times 10^2$ | $1.49 \times 10^2$ | $3.15 \times 10^2$ |

[Figure]

**Figure S16.** Cluster formation rate $J$ **(a)** and enhancement strength $R$ of MSA **(b)** as a function of $[MSA] = 10^6 – 10^8$ molec. cm$^{-3}$, with different energy of $\Delta G_{278K}$ (black line), $\Delta G_{278K} + 1$ (blue line), $\Delta G_{278K} – 1$ (red line), at $T = 278$ K, CS = 2.0 × 10$^{-3}$ s$^{-1}$, [HIO$_3$] = 10$^7$, [HIO$_2$] = 2.0 × 10$^5$ molec. cm$^{-3}$.

For the convenience of the review, we have copied Table S8 and Figure S16 and the corresponding analysis (in the revised supporting file) as following: "As reported by Kupiainen[9] et al. (2012), the differences between the computational (DFT//RI-CC2 method) and experimental $\Delta G$ values are about 1 kcal mol$^{-1}$ or less[10]. Accordingly, Almedia[11] et al. (2013) calculated the uncertainty range of ACDC simulated cluster formation resulting from QC calculations by adjusting the binding energy (±1 kcal mol$^{-1}$). Further given the consistency of our research framework (DFT//RI-CC2 + ACDC) with Almedia et al. (2013), herein we have performed the uncertainty analysis of $R_{MSA}$ caused by QC calculations through adding or subtracting 1 kcal mol$^{-1}$ from the $\Delta G$ (using $\Delta G_{278K}$ as a reference). As shown in Table S8 and Fig. S16, adjusting the $\Delta G_{278K}$ of clusters by ±1 kcal mol$^{-1}$ resulted in a minor variation in $J$ and $R$ of MSA, with the overall trend remaining consistent."

\------------------------

**Technical comments:**

**Comment 5:** Line 24: nucleating -> nucleation. This replacement should be done in several places in the text.

**Response:** Thanks for this helpful comment., the "nucleating" has been changed to "nucleation" in the revised manuscript.

\-\-\-\-\-\-\-\-\-\-\-\-\-\-\-\-\-\-\-\-\-\-\-\-\-

**Comment 6:** Line 27: remove the second comma.

**Response:** The second comma has been removed in the revised manuscript.

\-\-\-\-\-\-\-\-\-\-\-\-\-\-\-\-\-\-\-\-\-\-\-\-\-

**Comment 7:** Line 35: might be rewritten as: Although the efficient nucleation of $HIO_3$ and $HIO_2$ is overall consistent with the CLOUD measurements, this mechanism does not account for all $HIO_3$-induced nucleation in the real atmosphere.

**Response:** Thanks, the wording suggested by the reviewer is more appropriate. Accordingly, the sentence has been rewritten as "Although the efficient nucleation of $HIO_3$ and $HIO_2$ is overall consistent with the CLOUD measurements, this mechanism does not account for all $HIO_3$-induced nucleation in the real atmosphere." in Lines 34-36 of the revised manuscript.

\-\-\-\-\-\-\-\-\-\-\-\-\-\-\-\-\-\-\-\-\-\-\-\-\-

**Comment 8:** Line 48: might be rewritten as: the importance of the $HIO_3$-$HIO_2$-MSA nucleating mechanism may differ under distinct ambient conditions.

**Response:** According to this helpful suggestion, the sentence has been rewritten as "the importance of the $HIO_3$-$HIO_2$-MSA nucleation mechanism may differ under distinct ambient conditions" in Lines 47-48 of the revised manuscript.

\-\-\-\-\-\-\-\-\-\-\-\-\-\-\-\-\-\-\-\-\-\-\-\-\-

**Comment 9:** Line 72: remove 'and'

**Response:** According to the reviewer's suggestion, the "and" has been removed in the Line 71 of revised manuscript.

\-\-\-\-\-\-\-\-\-\-\-\-\-\-\-\-\-\-\-\-\-\-\-\-\-

**Comment 10:** Line 116: present -> presented

**Response:** The "present" has been corrected as "presented" in the Line 122 of revised manuscript.

\-\-\-\-\-\-\-\-\-\-\-\-\-\-\-\-\-\-\-\-\-\-\-\-\-

**Comment 11:** Line 144: observed -> shown

**Response:** The "observed" has been corrected as "shown" in the Line 153 of revised manuscript.

\-\-\-\-\-\-\-\-\-\-\-\-\-\-\-\-\-\-\-\-\-\-\-\-\-

**Comment 12:** Line 160: across-> through

**Response:** The "across" has been corrected as "through" in the Line 162 of revised manuscript.
* * *
**Comment 13:** Line 169: contribute to 74% of cluster formation

**Response:** Accordingly, the sentence has been corrected as "contribute to 74% of cluster formation" in the Line 171 of revised manuscript.
* * *
**Comment 14:** Line 214: access-> assess

**Response:** Thanks for the reviewer's careful reading, the "access" has been corrected as "assess" in the Line 240 of revised manuscript.

**Reference:**

[1] M.J. McGrath, T. Olenius, I.K. Ortega, V. Loukonen, P. Paasonen, T. Kurtén, M. Kulmala, H. Vehkamäki, Atmospheric Cluster Dynamics Code: a flexible method for solution of the birth-death equations, Atmos. Chem. Phys. 12 (2012) 2345–2355.

[2] K.-M. Oona, O. Tinja, Atmospheric Cluster Dynamics Code Technical manual, https://github.com/tolenius/ACDC/blob/main/ACDC_Manual_2020_11_25.pdf (2020).

[3] L.J. Beck, N. Sarnela, H. Junninen, C.J.M. Hoppe, O. Garmash, F. Bianchi, M. Riva, C. Rose, O. Peräkylä, D. Wimmer, O. Kausiala, T. Jokinen, L. Ahonen, J. Mikkilä, J. Hakala, X.C. He, J. Kontkanen, K.K.E. Wolf, D. Cappelletti, M. Mazzola, R. Traversi, C. Petroselli, A.P. Viola, V. Vitale, R. Lange, A. Massling, J.K. Nøjgaard, R. Krejci, L. Karlsson, P. Zieger, S. Jang, K. Lee, V. Vakkari, J. Lampilahti, R.C. Thakur, K. Leino, J. Kangasluoma, E.M. Duplissy, E. Siivola, M. Marbouti, Y.J. Tham, A. Saiz-Lopez, T. Petäjä, M. Ehn, D.R. Worsnop, H. Skov, M. Kulmala, V.M. Kerminen, M. Sipilä, Differing Mechanisms of New Particle Formation at Two Arctic Sites, Geophys. Res. Lett. 48 (2021).

[4] L.L.J. Quéléver, L. Dada, E. Asmi, J. Lampilahti, T. Chan, J.E. Ferrara, G.E. Copes, G. Pérez-Fogwill, L. Barreira, M. Aurela, D.R. Worsnop, T. Jokinen, M. Sipilä, Investigation of new particle formation mechanisms and aerosol processes at Marambio Station, Antarctic Peninsula, Atmos. Chem. Phys. 22 (2022) 8417–8437.

[5] M. Kulmala, T. Petäjä, T. Nieminen, M. Sipilä, H.E. Manninen, K. Lehtipalo, M. Dal Maso, P.P. Aalto, H. Junninen, P. Paasonen, I. Riipinen, K.E.J. Lehtinen, A. Laaksonen, V.-M. Kerminen, Measurement of the nucleation of atmospheric aerosol particles, Nature Protocols 7 (2012) 1651-1667.

[6] D. Xia, J. Chen, H. Yu, H.B. Xie, Y. Wang, Z. Wang, T. Xu, D.T. Allen, Formation Mechanisms of Iodine-Ammonia Clusters in Polluted Coastal Areas Unveiled by Thermodynamics and Kinetic Simulations, Environ. Sci. Technol. 54 (2020) 9235-9242.

[7] H. Yu, L. Ren, X. Huang, M. Xie, J. He, H. Xiao, Iodine speciation and size distribution in ambient aerosols at a coastal new particle formation hotspot in China, Atmos. Chem. Phys. 19 (2019) 4025–4039.

[8] M. Sipilä, N. Sarnela, T. Jokinen, H. Henschel, H. Junninen, J. Kontkanen, S. Richters, J. Kangasluoma, A. Franchin, O. peräkylä, M.P. Rissanen, M. Ehn, H. Vehkamäki, T. Kurten, T. Berndt, T. Petäjä, D. Worsnop, D. Ceburnis, V.M. Kerminen, M. Kulmala, C. O'Dowd, Molecular-scale evidence of aerosol particle formation via sequential addition of $HIO_3$, Nature 537 (2016) 532–534.

[9] O. Kupiainen, I.K. Ortega, T. Kurtén, H. Vehkamäki, Amine substitution into sulfuric acid – ammonia clusters, Atmos. Chem. Phys. 12 (2012) 3591-3599.

[10] K.D. Froyd, E.R. Lovejoy, Bond energies and structures of ammonia-sulfuric acid positive cluster ions, J. Phys. Chem. A 116 (2012) 5886-5899.

[11] J. Almeida, S. Schobesberger, A. Kürten, I.K. Ortega, O. Kupiainen-Määttä, A.P. Praplan, A. Adamov, A. Amorim, F. Bianchi, M. Breitenlechner, A. David, J. Dommen, N.M. Donahue, A. Downard, E. Dunne, J. Duplissy, S. Ehrhart, R.C. Flagan, A. Franchin, R. Guida, J. Hakala, A. Hansel, M. Heinritzi, H. Henschel, T. Jokinen, H. Junninen, M. Kajos, J. Kangasluoma, H. Keskinen, A. Kupc, T. Kurten, A.N. Kvashin, A. Laaksonen, K. Lehtipalo, M. Leiminger, J. Leppä, V. Loukonen, V. Makhmutov, S. Mathot, M.J. McGrath, T. Nieminen, T. Olenius, A. Onnela, T. Petäjä, F. Riccobono, I. Riipinen, M. Rissanen, L. Rondo, T. Ruuskanen, F.D. Santos, N. Sarnela, S. Schallhart, R. Schnitzhofer, J.H. Seinfeld, M. Simon, M. Sipilä, Y. Stozhkov, F. Stratmann, A. Tomé, J. Tröstl, G. Tsagkogeorgas, P. Vaattovaara, Y. Viisanen, A. Virtanen, A. Vrtala, P.E. Wagner, E. Weingartner, H. Wex, C. Williamson, D. Wimmer, P. Ye, T. Yli-Juuti, K.S. Carslaw, M. Kulmala, J. Curtius, U. Baltensperger, D.R. Worsnop, H. Vehkamäki, J. Kirkby, Molecular understanding of sulphuric acid-amine particle nucleation in the atmosphere, Nature 502

(2013) 359–363.

---

## Author Response (AR2)

Dear Editor,

Thanks sincerely for your handling our manuscript "**Molecular-level study on the role of methanesulfonic acid in iodine oxoacids nucleation**" (MS No.: egusphere-2023-2084). According to reviewer's professional and helpful comments, we have further revised the manuscript carefully. Additionally, we have included Dr. Biwu Chu in the author list, owing to his significant contribution in the uncertainty analysis section. The point-to-point responses to the Referee #4's comments are listed as below:

**Referee comments:**

The authors have addressed most of my concerns, but minor revision is still required before the manuscript can be published.

**Response:** We appreciate the valuable suggestions and positive feedback provided by the reviewer, which is helpful for further refining our manuscript and even holds significant guidance for our future research endeavors.

**Comments:**

1. The authors have made a mistake in connecting J1.2 and J1.5. When the cited formula is used, CS should have the unit of m^-2 instead of s^-1 (i.e., the value 0.002 s-1 should be converted to an area). Please refer to this paper: Analytical formulae connecting the "real" and the "apparent" nucleation rate and the nuclei number concentration for atmospheric nucleation events.

**Response:** This is a key and helpful point – thanks for bringing it up. This expert advice is beneficial for improving the accuracy of comparing the ACDC simulation with the field observations. According to the study of Kerminena and Kulmalab (2002), we have converted the CS value to CS' by the following equation (Kerminena and Kulmalab 2002):

$$CS = 4\pi D_i CS'$$

where $D_i$ is the diffusion coefficient of the condensing vapor, usually assumed to be sulfuric

acid (0.08 cm$^{-2}$ s$^{-1}$) (Kulmala et al. 2012). When the CS value is 0.002 s$^{-1}$, the CS' value is 18 m$^{-2}$.

Therefore, the relationship between the formation rates of simulated clusters ($J_{1.2}$) and observed clusters ($J_{1.5}$) can be written as (Kulmala et al. 2012):

$$J_{1.2} = J_{1.5} \exp\left\{0.23\times\left(\frac{1}{1.2}-\frac{1}{1.5}\right)\frac{CS'}{GR}\right\},$$

where GR was measured to be 3.2 – 4.4 nm·h$^{-1}$ in the 1.1 – 2.0 nm size range during three observed events, and CS' was 18 m$^{-2}$. Accordingly, the calculated $J_{1.2}$ is 1.1 – 1.2 times of $J_{1.5}$. The observed cluster formation rates for 1.5 nm clusters can be converted to the simulated $J_{1.2}$.

Thanks again for the reviewer's professional review. We have revised all the relevant figures (Figs. 6 and S14) and text (Lines 254, 256, 260 and 265) in the revised manuscript and supporting file.

2.  In performing the uncertainty analysis with respect to deltaG, it is my understanding that the authors have added 1 kcal/mol to (or subtracted 1 kcal/mol from) the calculated deltaG of all clusters. I'm not sure if the values of all cluster energies should move in the same direction. If this is indeed the case, can the authors explain why?

An alternate method to calculate the overall uncertainties is to do a Monte Carlo uncertainty analysis by doing many calculations. In each calculation, the deltaG of each cluster is randomly assigned a value within an uncertainty range. With J values from many of such calculations, an overall uncertainty can be derived.

**Response:** Thanks for this insightful suggestion from the reviewer, which is important for refining the uncertainty analysis. Indeed, as the reviewer mentioned, the potential deltaG energy bias of all clusters may not move towards the same direction. Hence, we have further performed a new uncertainty analysis based on the reviewer's advice. Accordingly, we have randomly assigned the $\Delta G$ bias of each cluster at 278 K in each calculation within the uncertainty range of -1 to 1 kcal mol$^{-1}$ (Almeida et al. 2013), then performed 500 ACDC calculations to obtain the cluster formation rates ($J$) based on the newly assigned $\Delta G$ values (Fig. S17).

For the convenience of the review, we have copied the added Fig. S17 and the corresponding analysis (in the revised supporting file) as following: "Furthermore, given that

the potential bias of all clusters may not move in the same direction, we have further randomly assigned $\Delta G$ value of each cluster ($\Delta G_{rand}$ at 278 K) within its uncertainty range ($\Delta G_{ref} - 1$ kcal mol$^{-1} < \Delta G_{rand} < \Delta G_{ref} + 1$ kcal mol$^{-1}$), where $\Delta G_{ref}$ indicates the results from the current quantum chemical calculations. Using the newly assigned $\Delta G_{rand}$, we have further performed the ACDC simulations to calculate the cluster formation rate ($J$). Figure S17 presents the results of 500 calculations. The results do not alter the overall trend of the rate variation, and the resulting principal conclusions of this study."

[Figure]

**Figure S17.** Cluster formation rate $J$ (cm$^{-3}$ s$^{-1}$) as a function of [MSA] = $10^6 - 10^8$ molec. cm$^{-3}$, with different energy of $\Delta G_{ref}$ (red line, reference condition), $\Delta G_{rand}$ (blue line, randomly assigned $\Delta G$ values of all clusters within the potential bias between -1 and 1 kcal mol$^{-1}$), at $T =$ 278 K, CS = $2.0 \times 10^{-3}$ s$^{-1}$, [HIO$_3$] = $10^7$, [HIO$_2$] = $2.0 \times 10^5$ molec. cm$^{-3}$.

**Minor:**

**1.** For the pseudo color plots (or heat map), the contour lines should be a bit thicker to improve their visibility.

**Response:** Thanks for the reviewer's careful reading. We have thickened the contour lines of all the heat maps in the revised manuscript (Fig. 5) and supporting file (Figs. S10 and S11).

2. Some of the wording in manuscript are a bit awkward. I suggest the authors do one more round of language editing.

**Response:** According to the reviewer's valuable comments, we have re-edited the language of the article. The specific modifications are as follows:

Lines 15-17: The sentence 'The results show that MSA can form stable molecular clusters with $HIO_3$ and $HIO_2$ jointly via hydrogen and halogen bonds, as well as electrostatic attraction after proton transfer to $HIO_2$' has been changed to 'Our results show that MSA can form stable molecular clusters with $HIO_3$ and $HIO_2$ jointed via hydrogen bond, halogen bond, and electrostatic attraction after proton transfer to $HIO_2$'.

Lines 18-20: The sentence 'Furthermore, adding MSA significantly enhance the rate of $HIO_3$-$HIO_2$-based cluster formation, even up to $10^4$-fold at cold marine regions with rich MSA and scarce iodine, such as polar Ny-Ålesund and Marambio' has been changed to 'Furthermore, our results show that considering MSA will significantly enhance the calculated rate of $HIO_3$-$HIO_2$-based cluster formation, with up to $10^4$-fold at cold marine regions containing rich MSA and scarce iodine, such as polar Ny-Ålesund and Marambio'.

Line 22: The word 'burst' has been changed to 'bursts'.

Lines 24-25: The sentence 'Marine aerosol, the primary natural aerosol, significantly impact global climate, radiation balance, and even human health' has been changed to 'Marine aerosol, which is the primary natural aerosol, has a significant impact on global climate, radiation balance, and even human health'.

Line 25: The word 'main' has been changed to 'primary'.

Line 27: The word 'And' has been changed to 'Moreover'.

Line 40: The word 'real' has been changed to 'authentic'.

Lines 84-85: The sentence 'To probe the binding nature within molecular clusters, wavefunction analysis was conducted using Multiwfn 3.7' has been changed to 'Wavefunction analysis was carried out using Multiwfn 3.7 to investigate the binding nature within molecular clusters'.

Line 108: The word 'are' has been changed to 'is'.

Line 109: Added the sentence '(Table S8 and Figs. S15, S16 and S17)'.

Line 130: The word 'greater' has been changed to 'stronger'.

Line 151: The word 'affect' has been changed to 'affects'.

Line 167: The sentence 'through the collision of' has been changed to 'by colliding'.

Lines 217-218: The sentence 'In fact, apart from atmospheric temperature, precursor concentrations may vary regionally or seasonally, further affecting nucleation' has been changed to 'It's worth noting that apart from atmospheric temperature, precursor concentrations might also vary regionally or seasonally, which can further affect nucleation'.

Line 283: The sentence 'Compared to previously the reported $HIO_3$-$HIO_2$ system' has been changed to 'Compared to the $HIO_3$-$HIO_2$ system reported previously'.

Lines 286-289: The sentence 'Further comparison with field observations indicates that the $HIO_3$-$HIO_2$-MSA synergistic nucleation plays a limited role in the mid-latitude oceans, particularly with abundant iodine (e.g., Mace Head), but an important role in the colder polar regions (e.g., Ny-Ålesund and Marambio)' has been changed to 'Further comparison with field observations indicates that the $HIO_3$-$HIO_2$-MSA synergistic nucleation plays a limited role in mid-latitude ocean regions, particularly in regions with abundant iodine (e.g., Mace Head), but a potential role in colder polar regions (e.g., Ny-Ålesund and Marambio)'.

Line 288: The sentence 'In addition to MSA, given the complex oceanic atmosphere, other potential nucleation precursors, such as sulfuric acid and amines, may also affect the $HIO_3$-$HIO_2$ nucleation process, further contributing to the formation of marine iodine particles, which deserves future studies' has been changed to 'Given the complex oceanic atmosphere, other potential nucleation precursors beyond MSA, such as sulfuric acid and amines, may also affect the $HIO_3$-$HIO_2$ nucleation process and further contribute to the formation of marine iodine particles, which deserves future investigations'.

Thanks again for the reviewers' professional and valuable suggestions, we have done our best to refine our manuscript.

**Reference:**

Kerminena, V.-M. & M. Kulmalab 2002 Analytical formulae connecting the "real" and the "apparent" nucleation rate and the nuclei number concentration for atmospheric nucleation events. *Aerosol Science,* 33**,** 609-622.

Kulmala, M., T. Petäjä, T. Nieminen, M. Sipilä, H. E. Manninen, K. Lehtipalo, M. Dal Maso, P. P. Aalto, H. Junninen, P. Paasonen, I. Riipinen, K. E. J. Lehtinen, A. Laaksonen & V.-M. Kerminen 2012 Measurement of the nucleation of atmospheric aerosol particles. *Nature Protocols,* 7**,** 1651-1667.

Almeida, J., S. Schobesberger, A. Kürten, I. K. Ortega, O. Kupiainen-Määttä, A. P. Praplan, A. Adamov, A. Amorim, F. Bianchi, M. Breitenlechner, A. David, J. Dommen, N. M. Donahue, A. Downard, E. Dunne, J. Duplissy, S. Ehrhart, R. C. Flagan, A. Franchin, R. Guida, J. Hakala, A. Hansel, M. Heinritzi, H. Henschel, T. Jokinen, H. Junninen, M. Kajos, J. Kangasluoma, H. Keskinen, A. Kupc, T. Kurtén, A. N. Kvashin, A. Laaksonen, K. Lehtipalo, M. Leiminger, J. Leppä, V. Loukonen, V. Makhmutov, S. Mathot, M. J. McGrath, T. Nieminen, T. Olenius, A. Onnela, T. Petäjä, F. Riccobono, I. Riipinen, M. Rissanen, L. Rondo, T. Ruuskanen, F. D. Santos, N. Sarnela, S. Schallhart, R. Schnitzhofer, J. H. Seinfeld, M. Simon, M. Sipilä, Y. Stozhkov, F. Stratmann, A. Tomé, J. Tröstl, G. Tsagkogeorgas, P. Vaattovaara, Y. Viisanen, A. Virtanen, A. Vrtala, P. E. Wagner, E. Weingartner, H. Wex, C. Williamson, D. Wimmer, P. Ye, T. Yli-Juuti, K. S. Carslaw, M. Kulmala, J. Curtius, U. Baltensperger, D. R. Worsnop, H. Vehkamäki & J. Kirkby 2013 Molecular understanding of sulphuric acid–amine particle nucleation in the atmosphere. *Nature,* 502**,** 359-363.